# ZMYND11-MBTD1 induces leukemogenesis through hijacking NuA4/TIP60 acetyltransferase complex and a PWWP-mediated chromatin association mechanism

Jie Li[1,2,3], Phillip M. Galbo Jr.[4], Weida Gong[1], Aaron J. Storey[5], Yi-Hsuan Tsai[1], Xufen Yu[6], Jeong Hyun Ahn[1,2], Yiran Guo[1,3], Samuel G. Mackintosh[5], Ricky D. Edmondson[5], Stephanie D. Byrum[5], Jason E. Farrar[7], Shenghui He[1,8], Ling Cai[1,8], Jian Jin[6], Alan J. Tackett[5,7], Deyou Zheng[4,9] & Gang Greg Wang[1,2,3✉]

Recurring chromosomal translocation t(10;17)(p15;q21) present in a subset of human acute myeloid leukemia (AML) patients creates an aberrant fusion gene termed ZMYND11-MBTD1 (ZM); however, its function remains undetermined. Here, we show that ZM confers primary murine hematopoietic stem/progenitor cells indefinite self-renewal capability ex vivo and causes AML in vivo. Genomics profilings reveal that ZM directly binds to and maintains high expression of pro-leukemic genes including Hoxa, Meis1, Myb, Myc and Sox4. Mechanistically, ZM recruits the NuA4/Tip60 histone acetyltransferase complex to cis-regulatory elements, sustaining an active chromatin state enriched in histone acetylation and devoid of repressive histone marks. Systematic mutagenesis of ZM demonstrates essential requirements of Tip60 interaction and an H3K36me3-binding PWWP (Pro-Trp-Trp-Pro) domain for oncogenesis. Inhibitor of histone acetylation-'reading' bromodomain proteins, which act downstream of ZM, is efficacious in treating ZM-induced AML. Collectively, this study demonstrates AML-causing effects of ZM, examines its gene-regulatory roles, and reports an attractive mechanism-guided therapeutic strategy.

[1] Lineberger Comprehensive Cancer Center, University of North Carolina at Chapel Hill School of Medicine, Chapel Hill, NC, USA. [2] Department of Biochemistry and Biophysics, University of North Carolina at Chapel Hill School of Medicine, Chapel Hill, NC, USA. [3] Curriculum in Genetics and Molecular Biology, University of North Carolina at Chapel Hill, Chapel Hill, NC, USA. [4] Department of Genetics, Albert Einstein College of Medicine, Bronx, NY, USA. [5] Department of Biochemistry and Molecular Biology, University of Arkansas for Medical Sciences, Little Rock, AR, USA. [6] Mount Sinai Center for Therapeutics Discovery, Departments of Pharmacological Sciences and Oncological Sciences, Tisch Cancer Institute, Icahn School of Medicine at Mount Sinai, New York, NY, USA. [7] Winthrop P. Rockefeller Cancer Institute, University of Arkansas for Medical Sciences and Arkansas Children's Research Institute, Little Rock, AR, USA. [8] Department of Genetics, University of North Carolina at Chapel Hill School of Medicine, Chapel Hill, NC, USA. [9] Department of Neurology and Department of Neuroscience, Albert Einstein College of Medicine, Bronx, NY, USA. ✉email: greg_wang@med.unc.edu

Post-translational modification (PTM) of histones provides a fundamental means for modulating gene expression and determining cellular identities during development and cell differentiation, and its deregulation is intimately associated with pathogenesis of human cancers, including acute myeloid leukemia (AML)[1–6]. Recently, a new recurrent chromosomal translocation t(10;17)(p15;q21) was detected among a subset of AML patients, which produces an abnormal chimeric gene by fusing an N-terminal gene segment (i.e., exons 1–11 or 1–12) of Zinc Finger MYND-Type Containing 11 (ZMYND11) in-frame with the entire coding region (exons 3–17) of Malignant Brain Tumor domain containing 1 (MBTD1)[7–11]. The resultant chimeric protein termed ZMYND11-MBTD1 (hereafter referred to as "ZM") harbors the PHD, Bromo and PWWP domains of ZMYND11 and full-length MBTD1 (Fig. 1a). In the clinic, ZM-positive AML cases often display a minimally differentiated cell phenotype and express poor prognosis-related gene markers (such as HOX cluster genes)[8,12], implicative of adverse outcome; however, the role for ZM in oncogenesis remains elusive.

ZMYND11, also known as BS69, was originally defined as a transcriptional co-repressor and putative tumor suppressor, due to its capability to directly interact with and inhibit the transactivation activities of a set of viral and cellular oncoproteins, which include Adenovirus Early Region 1A (E1A), Epstein–Barr virus nuclear antigen 2 (EBNA2) and MYB[13–15]. Such inhibition relies on interaction between a common PXLXP motif present in E1A, EBNA2 or MYB and ZMYND11's MYND domain, a C-terminal module lost in the AML-associated ZM chimera; in addition, ZMYND11's MYND domain recruits additional repressors such as Nuclear Receptor Corepressor (N-CoR) and histone deacetylase (HDAC) complexes[16,17]. While MYND provides an interface for various protein-protein interactions, ZMYND11's N-terminal domains (PHD, Bromo and PWWP) contain the chromatin-association activities and are retained in ZM— notably, a Bromo-PWWP tandem module was recently shown to function as a 'reader' specific for tri-methylated histone H3.3 lysine-36 (H3.3K36me3), thereby targeting ZMYND11 to actively transcribed genes[18,19]. On the other hand, MBTD1 consists of a FCS-type zinc finger (ZnF) at the N-terminus and four Malignant Brain Tumor (MBT) repeats at the C-terminus, with its fourth MBT domain (MBT4) harboring a binding activity for mono and di-methylated histone H4 lysine-20 (H4K20me1/2)[20]. Recently, MBTD1 was identified as a subunit of the NuA4/TIP60 acetyltransferase complex, a potent transcriptional coactivator through its histone-acetylation activity[21,22]. Interestingly, in addition to ZM seen in AML patients, several components of the NuA4/TIP60 complex including MBTD1 were frequently involved in aberrant chromosomal rearrangements detected in endometrial stromal sarcomas, highlighting a broader role for this group of chromatin regulators in oncogenesis[23,24].

In this study, we assess putative leukemogenic functions of ZM and dissect the underlying oncogenic mechanisms, aiming to develop mechanism-based therapeutic interventions for the affected AML patients.

## Results

**ZMYND11-MBTD1 (ZM) promotes proliferation of primary hematopoietic stem/progenitor cells (HSPCs) and arrests their terminal differentiation in vitro.** To assess the role for aberrant ZM chimera in leukemogenesis, we retrovirally transduced ZM, either full-length (WT) or with either of the two fusion partners truncated (i.e., ΔZMYND11 or ΔMBTD1, Fig. 1a, b), into HSPCs isolated from mouse bone marrow, followed by monitoring of cell proliferation and differentiation. In independently performed experiments, only HSPCs transduced with WT ZM (tagged by either 3xHA3xFlag or GFP, Fig. 1b and Supplementary Fig. 1a) propagated as progenitors indefinitely (for over three months; data not shown) and became immortalized in the presence of supporting cytokines (Fig. 1c, d). In contrast, those transduced with empty vector control (EV) or a ZMYND11- or MBTD1-truncated ZM proliferated transiently (Fig. 1c) and were then subject to terminal differentiation (data not shown). Fluorescence-activated cell sorting (FACS) analysis of cultures two weeks post-transduction revealed that only WT ZM, and not EV or ZM's fusion partner truncated versions, was able to sustain a myeloid stem/progenitor cell subpopulation (c-Kit+/Cd34+/Mac1+; Supplementary Fig. 1b). We have also used a semi-solid culture system to assess transforming capabilities of ZM, and consistent to what was observed with the liquid culture system, only WT ZM and not EV or the two partner-truncated forms displayed a robust promoting effect on colony formation (Fig. 1e, f and Supplementary Fig. 1c). FACS of multiple lines immortalized by WT ZM consistently revealed an AML progenitor phenotype (c-Kit+/Cd34+/Mac1+/Cd19-/Cd4-/Cd8-; Fig. 1g and Supplementary Fig. 1d, e). Given that cellular origins of leukemia remarkably influence clinical outcomes and therapy response[25,26], we also examined which cell type in the Lin- HSPC population is susceptible to transformation by ZM. To this end, we purified murine HSC (Lin-/cKit+/Sca1+/Cd16/32-/Cd150+/Cd48-), GMP (Lin-/cKit+/Sca1-/Cd16/32+/Cd150-) and differentiated myeloid cells (cKit-/Mac1+) from bone marrow (Supplementary Fig. 1f). Following its transduction, ZM conferred both HSC and GMP, but not differentiated myeloid cells, the capacity of in vitro immortalization, with the HSC-derived immortalized cells proliferating slightly faster than those GMP-derived ones (Supplementary Fig. 1g), and these two ZM-immortalized leukemia lines exhibited similar myeloid blast immunophenotype (cKit+/Cd34+/Mac1+) and morphology (Supplementary Fig. 1h, i). Together, we demonstrate that human AML-associated ZM chimera is able to promote proliferation and arrest terminal differentiation of murine HSPCs (including the purified HSC and GMP), inducing their immortalization in vitro.

**ZM induces AML in mouse models.** We next asked whether ZM is capable of causing AML in vivo. Towards this end, we transplanted murine HSPCs freshly infected with retrovirus expressing either EV or ZM into a cohort of sub-lethally irradiated syngeneic mice. While the control mice remained healthy over a 360-day monitoring period, those transplanted with ZM-transduced HSPCs gradually developed and succumbed to leukemia, with an average latency of 154 days (8 out of 9 mice; Fig. 1h, ZM versus EV)– during disease progression, there was a dramatic expansion of ZM-transduced HSPCs (luciferase-labeled) in the bone and spleen (Fig. 1i); as a result, leukemic mice displayed elevated counts of white blood cells (WBCs) in peripheral blood (Fig. 1j, k) and splenomegaly (Fig. 1l, m) at their terminal stage. Pathological analyses also showed that, compared to controls, spleens from the ZM-transplanted cohort displayed massive infiltration of leukemic blasts and a concomitant disruption of normal splenic structure (Fig. 1n versus 1o). ZM-induced phenotypes were found to be typical of AML— the bone marrow and spleen of leukemic mice were filled with myeloblasts (Fig. 1p, q) that expressed the exogenously introduced ZM fusion (Fig. 1r, top panel) and the surface markers of immature myeloid progenitors (Fig. 1s, cKit+Mac1+).

A 'two-hit' model has been widely accepted for AML pathogenesis[27], in which full-blown and more aggressive AMLs are frequently caused by combinatorial genetic mutations, with one perturbing hematopoietic cell self-renewal or differentiation (often a mutation of a transcription/epigenetic factor gene) and the other

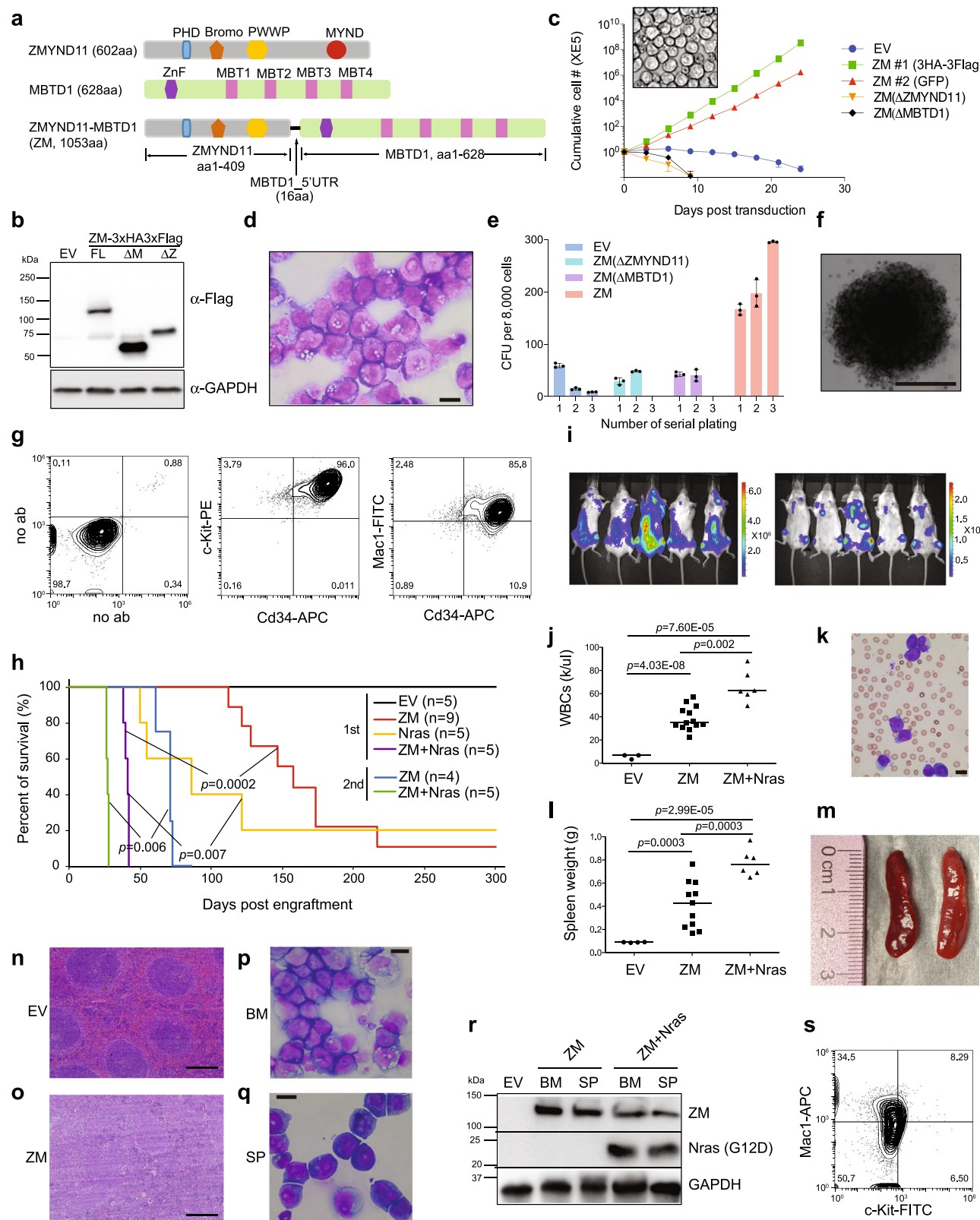

promoting proliferation (often a mutation of a Receptor Tyrosine Kinase [RTK] signal transduction-related protein). We thus reasoned that ZM, a lesion targeting chromatin regulator, likely cooperates with a RTK pathway-related mutation during AML development. Indeed, co-expression of an AML-associated, constitutively active NrasG12D mutant significantly promoted kinetics of ZM-induced AMLs (Fig. 1h, with an average latency of 41 days

for the "ZM + Nras" cohort)—during disease progression, leukemia cells co-expressing ZM and NrasG12D (luciferase-labeled) homed to and expanded in bones and spleen (Fig. 1r and Supplementary Fig. 2a). AMLs caused by co-expressed ZM and NrasG12D also displayed typical leukemic phenotypes including elevated counts of circulating WBCs (Fig. 1j and Supplementary Fig. 2b), splenomegaly (Fig. 1l and Supplementary Fig. 2c) and massive expansion of

**Fig. 1 ZMYND11-MBTD1 (ZM), a human AML-associated chimeric gene, induces immortalization of murine HSPCs in vitro and causes AML in vivo.**
**a** Scheme showing domain structures of ZMYND11, MBTD1 and the AML-associated ZM fusion. **b** Immunoblotting for ZM, either full-length (WT) or with its ZMYND11 or MBTD1 segment deleted (ΔZ or ΔM), post-transduction into murine HSPCs. **c** Proliferation kinetics of murine HSPCs transduced with empty vector (EV) or the indicated ZM (either WT, ΔZ or ΔM) in liquid culture (n = 3 biological replicates per group; data presented as mean ± SD). Insert shows a typical light micrograph of WT ZM-immortalized progenitors (scale bar = 10 um). **d** Wright-Giemsa staining of WT ZM-immortalized progenitors (scale bar = 10 um). **e** Quantification of colony-forming units (CFU, n = 3 biological replicates per group, with data presented as mean ± SD) formed by HSPCs stably transduced with either EV or indicated ZM. **f** Representative colony of HSPCs expressing WT ZM at the third replating of CFU assays (scale bar = 1 mm). **g** FACS of WT ZM-immortalized progenitors. **h** Kaplan–Meier survival curve of syngeneic mice post-transplantation of either HSPCs transduced by EV or the indicated gene (in the first injection group) or bone marrow cells isolated from leukemic mice (in the secondary injection group). n = cohort size; the p values were calculated by two-sided log-rank test. **i** Bioluminescence images of mice, with dorsal and ventral views shown on the left and right respectively, 8 weeks post-transplantation of HSPCs coexpressing ZM and a luciferase reporter. **j, k** Summary of white blood cell counts (WBC; **j**) and a representative Wright-Giemsa staining image of blood smear (ZM alone, **k**; scale bar = 10 μm) using peripheral blood prepared from recipient mice at their terminal stage of AMLs induced by either ZM alone (middle) or coexpression of ZM plus Nras$^{G12D}$ (right), relative to mock-treated controls (left, EV). The p values (panel **j**; n = 3, 13, and 6 mice for EV, ZM and ZM + Nras group, respectively) were calculated by two-sided Student's t test. **l, m** Summary of spleen weight (**l**) and image showing enlarged spleens (ZM alone, **m**) of leukemic mice induced by either ZM alone (middle) or coexpression of ZM plus Nras$^{G12D}$(right), relative to mock (left). The p values (panel **l**; n = 4, 11, and 6 mice for EV, ZM and ZM + Nras group, respectively) were calculated by two-sided Student's t test. **n, o** H&E staining of spleen from the indicated cohort, either mock-treated (EV, **n**) or with ZM-induced AML (**o**), scale bar = 200 um. **p, q** Wright-Giemsa staining of cells derived from bone marrow (BM, **p**) and spleen (SP, **q**) from mice in the ZM cohort that developed full-blown AML (scale bar = 10 um). **r** Immunoblotting for ZM (anti-Flag, top) and Nras$^{G12D}$ mutant (anti-Flag, middle) in the bone marrow-and spleen-derived leukemic cells isolated from the indicated cohort. **s** FACS of ZM-induced primary AML cells isolated from bone marrow.

AML blasts in bone marrow and spleen (Cd34+/Mac1+; Supplementary Fig. 2d–f). Finally, we performed the secondary transplantation and found that mice receiving primary tumor cells, which were freshly isolated from bone marrow of leukemic mice, developed AML with a significantly shorter latency (Fig. 1h), demonstrating that ZM-induced disease is indeed AML.

Additionally, considering that ZM alone caused murine AML only after a long latency with incomplete penetrance, we speculated that additional cooperating mutations that occur spontaneously during AML clonal evolution underlie ZM-induced AML development. We thus performed whole exome sequencing (WES) of primary murine AMLs and found that, in comparison with the original ZM-transduced HSPCs used for transplantation, the developed AML tumors indeed acquired somatic nonsynonymous mutations (Supplementary Data 1). Of note, one murine AML acquired a high-frequency missense mutation S502L in Protein tyrosine phosphatase non-receptor type 11 (Ptpn11; also known as Shp2) (Supplementary Fig. 2g). S502L and a similar S502P mutation of PTPN11/SHP2 were previously reported among human patients with Noonan syndrome or leukemia[28–31] and predicted to be pathogenic through locking PTPN11 in an active conformation, thereby leading to sustained activation of RAS/ MAPK cascade[32]. Thus, akin to co-introduced Nras$^{G12D}$, such spontaneous acquisition of a Ptpn11 activating mutation in ZM alone-initiated AML again highlights an oncogenic collaboration between ZM and a hyper-activated RAS signaling. Moreover, another examined murine AML was found to carry missense mutations in Sntg1 and Clca3a2, functional consequences of which are unclear and merit further studies.

Collectively, we show that ZM itself is sufficient to initiate and cause AML in mice, a disease accelerated by activated Ras signaling.

**RNA sequencing (RNA-seq) revealed that ZM maintains a gene-expression program related to stemness and leukemic transformation.** Next, to dissect the mechanisms underlying ZM-mediated leukemogenesis, we performed transcriptome analysis of ZM-transformed AML cells. Given a poorly differentiated AML phenotype associated with ZM in patients[8] and our animal models, we asked whether ZM activates the 'stemness' gene-expression program. Here, using the public datasets, we first defined a signature that contains 922 genes showing specific expression in primitive self-renewing HSPCs (Lin-/Sca1+/c-Kit+

or LSK cells) and downregulation among terminally differentiated blood cell lineages (Supplementary Data 2). Gene set enrichment analysis (GSEA) with a different RNA-seq dataset verified an expected unique enrichment of this 922-gene 'stemness'-related signature in primitive HSPCs, relative to differentiated hematological cells (Supplementary Fig. 3a). With the same method, we found 1672 transcripts to be upregulated in ZM-transformed AML cells relative to differentiated hematological cells, and remarkably, 259 (~15%) of them are 'stemness'-related genes, which include a set of transcription factors (TFs) such as Hoxa cluster, Meis1, Mn1, Prdm5, Gata2, Sox4, Myb, and Myc (Fig. 2a and Supplementary Fig. 3b; Supplementary Data 2). Given that some of these stemness TFs were previously reported to promote leukemogenesis, ZM is likely to sustain aberrant self-renewal and induce AML by maintaining a subset of stemness gene-expression program.

We also profiled global transcriptomes of two additional murine AML lines established by either MLL rearrangement (MLL-AF9) or co-expressed Hoxa9 plus Meis1 (A9M); MLL rearrangement is known to sustain stemness genes while co-expressed A9M is able to arrest differentiation at a myeloid progenitor stage[6,33]. Hierarchical clustering analysis showed a closer similarity of ZM+ AML cells to MLL-AF9+ ones, when compared with A9M+ AML cells or various differentiated hematopoietic cell lineages based on their expression of stemness genes (Supplementary Fig. 3c). Overall, ZM+ and MLL-AF9+ AML cells expressed more 'stemness' genes at higher levels than A9M+ cells (Supplementary Fig. 3c). Consistently, GSEA revealed that gene sets associated with HSCs and NPM1c+ AMLs were significantly enriched in ZM+ relative to A9M+ cells (Fig. 2b); conversely, myeloid differentiation-associated genes showed decreased expression in ZM+ relative to A9M+ cells, implying that, compared to A9M, ZM is able to arrest differentiation of HSPCs at a significantly earlier stage. We next determined differentially expressed genes (DEGs) between ZM+ and A9M+ AML cells (Fig. 2c and Supplementary Data 3). Consistent with GSEA results, DAVID (Database for Annotation, Visualization and Integrated Discovery) functional annotation of DEGs linked the ZM-activated transcripts to those related to homeobox proteins, transcriptional regulation and Wnt, whereas the ZM-suppressed ones exhibited significant enrichments in myeloid differentiation-related genes such as those involved in immune response or cell adhesion (Fig. 2d). In addition to A9M+ cells, we

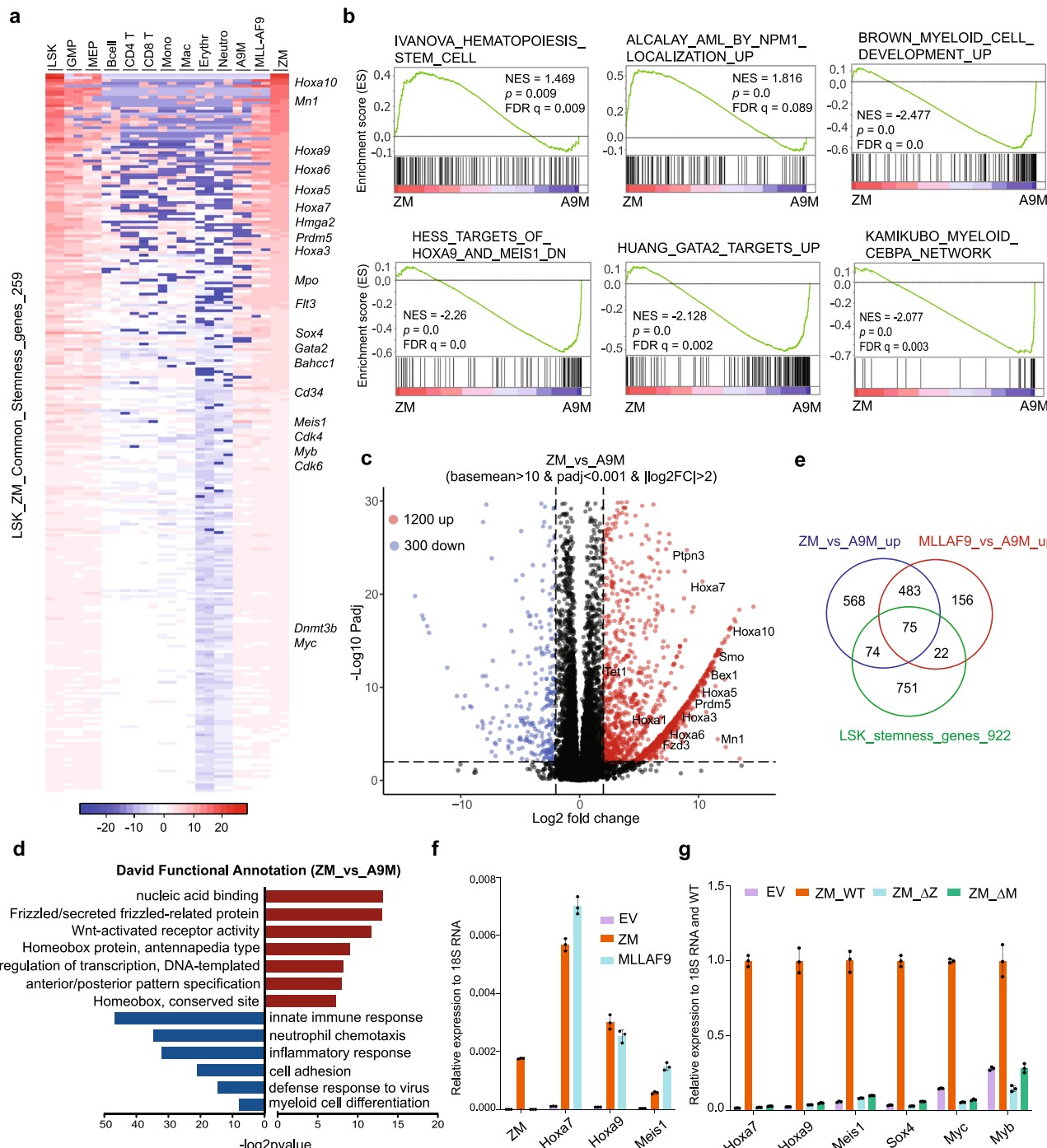

**Fig. 2 ZM enforces aberrant activation of a pro-leukemic stemness gene-expression program including Hoxa, Meis1, Sox4, Myc and Myb. a** Heatmap of 259 genes showing higher expression in both normal LSK (Lin-/Sca1+/c-Kit+) and ZM-immortalized murine AML cells, relative to mature blood cell types. Expression values represent mean-centered log2(TPM). Genes are sorted in a descending order based on their expression values in ZM+ AML cells. A9M and MLL-AF9 represent the murine AML lines established by the coexpressed Hoxa9 plus Meis1a (A9M) and MLL-AF9, respectively. **b** GSEA shows enrichment of the HSC-related and mutant-NPM1-upregulated genes in ZM+ relative to A9M+ cells, as well as enrichment of genes upregulated upon myeloid differentiation or the indicated TF-related transcripts in A9M+ relative to ZM+ cells. The p value (n = 15,282 genes per group) was calculated by an empirical phenotype-based permutation test. The FDR is adjusted for gene set size and multiple hypotheses testing while the p value is not. **c** Volcano plot displays differentially expressed genes (DEGs) identified in ZM+ versus A9M+ AML cells. The up- and down-regulated DEGs, defined with the indicated cutoff, are represented by red and blue dots, respectively, with example stemness-related genes highlighted. **d** DAVID functional annotation of DEGs identified in ZM+ versus A9M+ AML cells, with the enriched terms for upregulated (n = 1200 genes) and downregulated (n = 300 genes) DEGs indicated by red bars and blue bars, respectively. The p value was calculated by Fisher's Exact test. **e** Venn diagram showing overlaps among ZM-upregulated genes (versus A9M), MLL-AF9-upregulated genes (relative to A9M), and LSK stemness genes. **f** RT-PCR of Hoxa7, Hoxa9 and Meis1 in ZM- and MLL-AF9-transformed AML cells. qPCR signals from three independent experiments were normalized to those of 18S RNA and presented as mean ± SD. **g** RT-PCR of AML-related proto-oncogenes in murine HSPCs post-transduction of EV or the indicated ZM. qPCR signals from three independent experiments were normalized to those of 18S RNA and then WT ZM and presented as mean ± SD.

also employed HSPC culture two weeks post-transduction of empty vector (EV; which is a mixed HSPC-derived cell population under differentiation) as a second control to identify ZM-regulated genes (Supplementary Fig. 3d)— essentially, DAVID and GSEA analyses revealed same enrichments of the above-mentioned 'stemness'-, AML- or homeobox TF-related transcripts in ZM+ cells, compared to EV-transduced controls (Supplementary Fig. 3e, f).

Of note, a set of pro-leukemic/stemness TFs such as Hoxa and Mn1 were consistently found to be upregulated in ZM+ cells, relative to two independent controls (A9M+ AML cells in Fig. 2c and EV-transduced cells in Supplementary Fig. 3d). These stemness TFs were also found among transcripts showing higher expression in both ZM+ and MLL-AF9+ AML cells, relative to A9M+ ones, indicating a leukemic pathway commonly activated by ZM and MLL rearrangement (Fig. 2e and Supplementary Data 3). RT-qPCR further verified comparable upregulation of Hoxa7, Hoxa9, and Meis1 by ZM or MLL-AF9 (Fig. 2f), an effect not seen with fusion partner-truncated mutants of ZM (Fig. 2g; ΔZMYND11 or ΔMBTD1).

Together, we show that ZM aberrantly potentiates a LSK 'stemness' gene-expression program, which includes a subset of pro-oncogenic TFs (i.e., Hoxa cluster genes, Meis1, Mn1, Sox4, Myc, and Myb).

**Integrated chromatin immunoprecipitation-sequencing (ChIP-seq) and RNA-seq analyses revealed a positive correlation between ZM binding and proto-oncogene activation**. ZM showed an exclusive nuclear localization in cells (Supplementary Fig. 4a); we thus performed ChIP-seq to map out ZM's genome-wide occupancy. Two independent AML lines, transformed by either 3xHA-3xFLAG- or GFP-tagged ZM, were used for ChIP with HA- or GFP-specific antibodies (Supplementary Fig. 4b), generating biological replicates of ChIP-seq signals that are highly correlated (Supplementary Fig. 4c, d). Taking the commonly called ZM peaks, we found a vast majority of them to be located within promoter-proximal regions (i.e., +/−2.5 kb from transcription start sites or TSS) and gene bodies (Fig. 3a); consistently, averaged ChIP-seq signals also showed ZM peaks at TSS-proximal regions extending along the transcription units (Fig. 3b and Supplementary Fig. 4c). Genomic Regions Enrichment of Annotations Tool (GREAT) analysis of ZM-bound peaks revealed their significant enrichments at genes involved in abnormal hematopoiesis and those occupied by PML-RARA fusion or upregulated by MLL rearrangement (Fig. 3c).

Next, to dissect impact of ZM binding on target gene transcription, we carried out ChIP-seq of H3K27me3 and H3K36me3 (Supplementary Fig. 4b), two antagonizing histone marks related to gene repression and activation respectively, followed by correlational analysis of AML cell ChIP-seq and RNA-seq profiles. First, we observed a significant overlap of ZM peaks with those of H3K36me3 but not H3K27me3 (Fig. 3d); consistently, unsupervised clustering of whole-genome promoter-proximal regions, based on their ZM binding pattern difference, produced four distinct groups, which clearly show that genes bound by ZM are also decorated by H3K36me3 and devoid of repressive H3K27me3 marks (Fig. 3e; clusters a1/2 and b1/2 vs. cluster c). Here, ZM peaks showing highest binding extend into H3K36me3-decorated gene bodies (Fig. 3e, cluster a1/2) and intriguingly, many 'stemness' TFs and proto-oncogenes that we defined to be ZM-upregulated (Fig. 2), such as Hoxa, Meis1, Myc, Myb and Sox4 (Fig. 3e, label on left), were within this cluster characterized by a rather 'broad' ZM-binding pattern (Fig. 3f and Supplementary Fig. 4e). In accordance with a positive correlation between ZM binding and transactivation-related histone PTMs, a

higher ZM occupancy is positively correlated with a higher level of gene transcription when relating AML cell ZM ChIP-seq dataset to its RNA-seq profiles, strongly arguing ZM as a potential gene activator (Fig. 3g and Supplementary Fig. 4f, g). Finally, using integrated ChIP-seq and RNA-seq analyses, we defined the gene signature directly activated by ZM in HSPCs (Fig. 3h and Supplementary Data 4), as well as the "stemness" signature directly activated by ZM (Supplementary Fig. 4h and Supplementary Data 4), which again include AML-related TFs–Hoxa, Meis1, Myc, Myb and Sox4.

Collectively, our genomics profilings strongly demonstrate that ZM occupancy at cis-regulatory elements is generally positively correlated with transactivation-related histone modifications (H3K36me3) and transcriptional activity, as exemplified by a set of AML proto-oncogenes.

**ZM recruits the NuA4/Tip60 histone acetyltransferase complex, establishing "super-enhancers" to drive proto-oncogene activation**. To further gain insight into the chromatin targeting and gene activation mechanisms underlying ZM-mediated leukemogenesis, we sought to identify ZM interactomes by employing a proximity labeling-based strategy, i.e. BioID[34,35] with ZM-transduced primary HSPCs (Fig. 4a and Supplementary Fig. 5a), followed by mass spectrometry-based proteomics analyses. Here, two independent experiments, carried out using a biotin ligase (BirA) fused to either ZM's N- or C-terminus, consistently identified a number of NuA4/Tip60 acetyltransferase complex subunits as the most significantly enriched hits, which include Kat5/Tip60, p400/EP400, Epc1, Epc2, Trrap, Dmap1, and Vps72 (Fig. 4a; Supplementary Fig. 5b and Supplementary Data 5). By co-immunoprecipitation (CoIP), we further verified that both full-length and ZMYND11-deleted forms of ZM, but not the MBTD1-deleted one, interacted with Tip60 (Supplementary Fig. 5c), showing that ZM oncoprotein has capability for binding Tip60 complex due to its MBTD1 part, consistent with recent report[21,22].

To further assess whether ZM and Tip60 complex co-localize among AML cell genome, we performed ChIP-seq for Tip60 and histone acetylations (H3K27ac and H4ac) with ZM-transformed AML cells. Both global views and unsupervised clustering analysis of ChIP-seq peaks around promoter-proximal regions revealed a remarkable genome-wide overlap of ZM with Tip60, H3K27ac and H4ac (Fig. 4b, c and Supplementary Fig. 5d, e). Given a 'super-enhancer'-like, broad distribution pattern seen with the called ZM peaks at certain genes (Fig. 3e, f), we reasoned that ZM might recruit Tip60 to establish 'super-enhancers', characterized by densely clustered H3K27ac and H4ac[36], for stimulating hyper-transcription of proto-oncogenes. Indeed, super-enhancer calling based on the highly enriched H3K27ac at regions beyond the immediate vicinity of TSS (+/−2.5 kb) identified 1050 super-enhancers, which not only exhibited significant overlaps with the 'super-enhancer'-like peaks of ZM and Tip60 (Fig. 4d–f and Supplementary Fig. 5f) but importantly conferred their target genes overall significantly higher expression than genes lacking super-enhancer (Fig. 4g). Again, super-enhancer peaks of ZM, Tip60 and histone acetylations were found associated with the ZM-activated proto-oncogenes such as Hoxa7/a9, Myb, Myc, Meis1 and Sox4 (Fig. 4d, e, h–j and Supplementary Fig. 5f–h).

Together, we show that ZM interacts with NuA4/Tip60 complex forming 'super-enhancer'-like broad bindings for its target gene hyper-transcription, which are typically seen at key oncogenic loci.

**ZM-mediated leukemic transformation relies on its ability to recruit the NuA4/Tip60 acetyltransferase complex to target genes**. To functionally assess whether interaction with

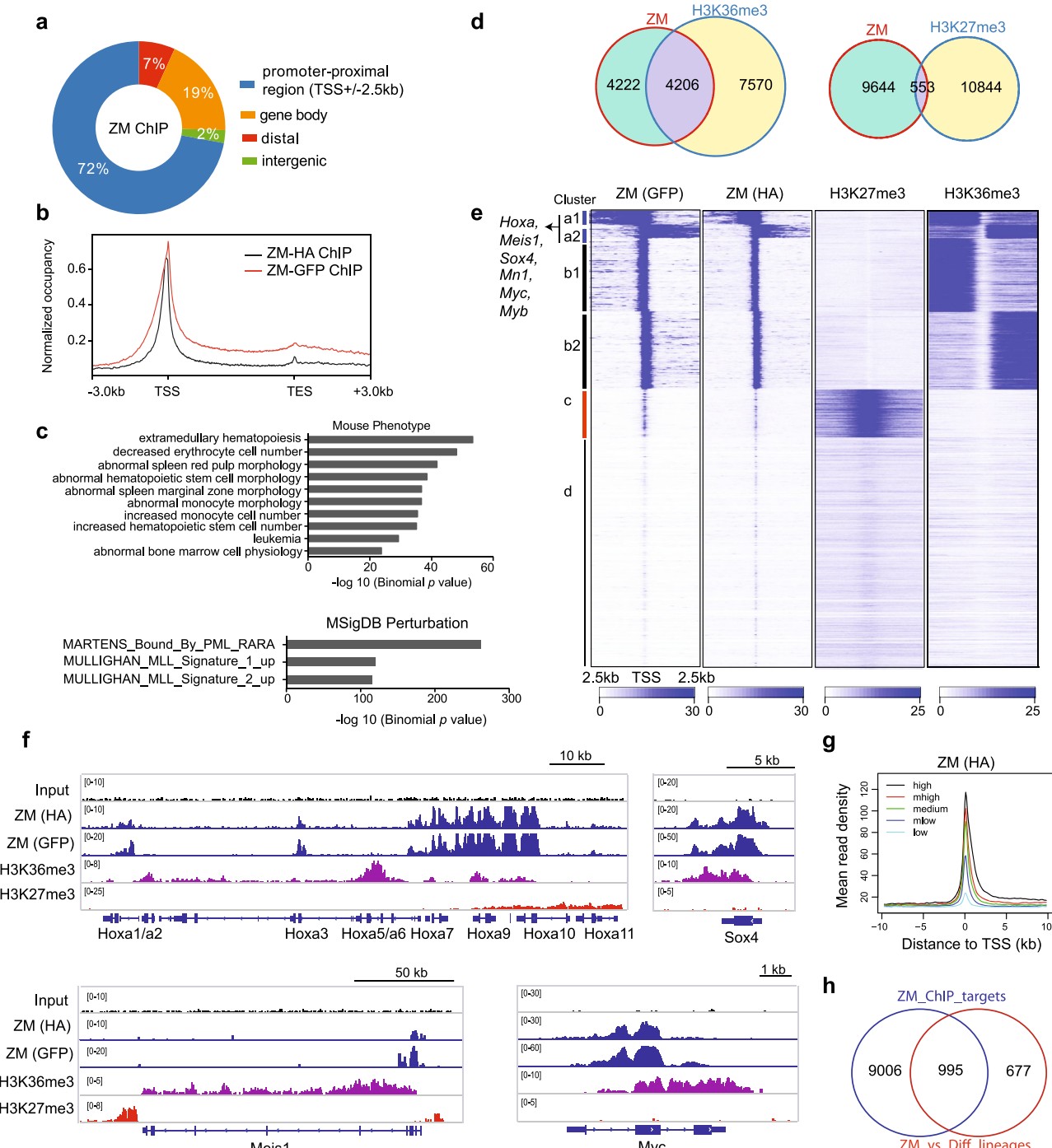

**Fig. 3 ChIP-seq reveals positive correlation between ZM binding and transcriptional activation of target genes, including oncogenic transcription factors (TFs) such as Hoxa, Meis1, Myc, Myb and Sox4. a** Genomic distribution of ZM peaks ($n = 16,114$; common to ZM HA and GFP ChIP) identified by ChIP-seq in ZM-transformed AML cells. **b** Average genome-wide ZM occupancy over transcription units, covering region from −3 kb upstream of transcriptional start site (TSS) to +3 kb downstream of transcriptional end site (TES). **c** Genomic Regions Enrichment of Annotations Tool (GREAT) analyses identifying enrichments of the indicated gene signatures among the called ZM peaks ($n = 16,114$ peaks). The $p$ value was calculated by binomial test. **d** Venn diagram displaying overlap between the indicated ChIP-seq peaks in ZM-transformed AML cells. **e** Heatmaps showing the indicated ChIP-seq read densities over promoter-proximal regions centered at TSS after k-means clustering. Four clusters (labeled as a to d) were produced based on their distinct ZM ChIP-seq signals. TSSs on the two DNA strands are labeled as 1 and 2 (such as a1/2 and b1/2), respectively. **f** IGV views of the indicated normalized ChIP-seq signals at AML-related proto-oncogene loci. The read counts were first normalized to 1x genome coverage (reads per genome coverage, RPGC) and then normalized to input. **g** Average ZM ChIP-seq signals at TSSs grouped by gene expression levels as revealed by RNA-seq, indicating a positive correlation. Genes were equally divided into 5 groups from high, medium high ("mhigh"), medium, medium low ("mlow"), to low expression. **h** Venn diagram showing overlap between ZM-activated genes and those directly bound by ZM.

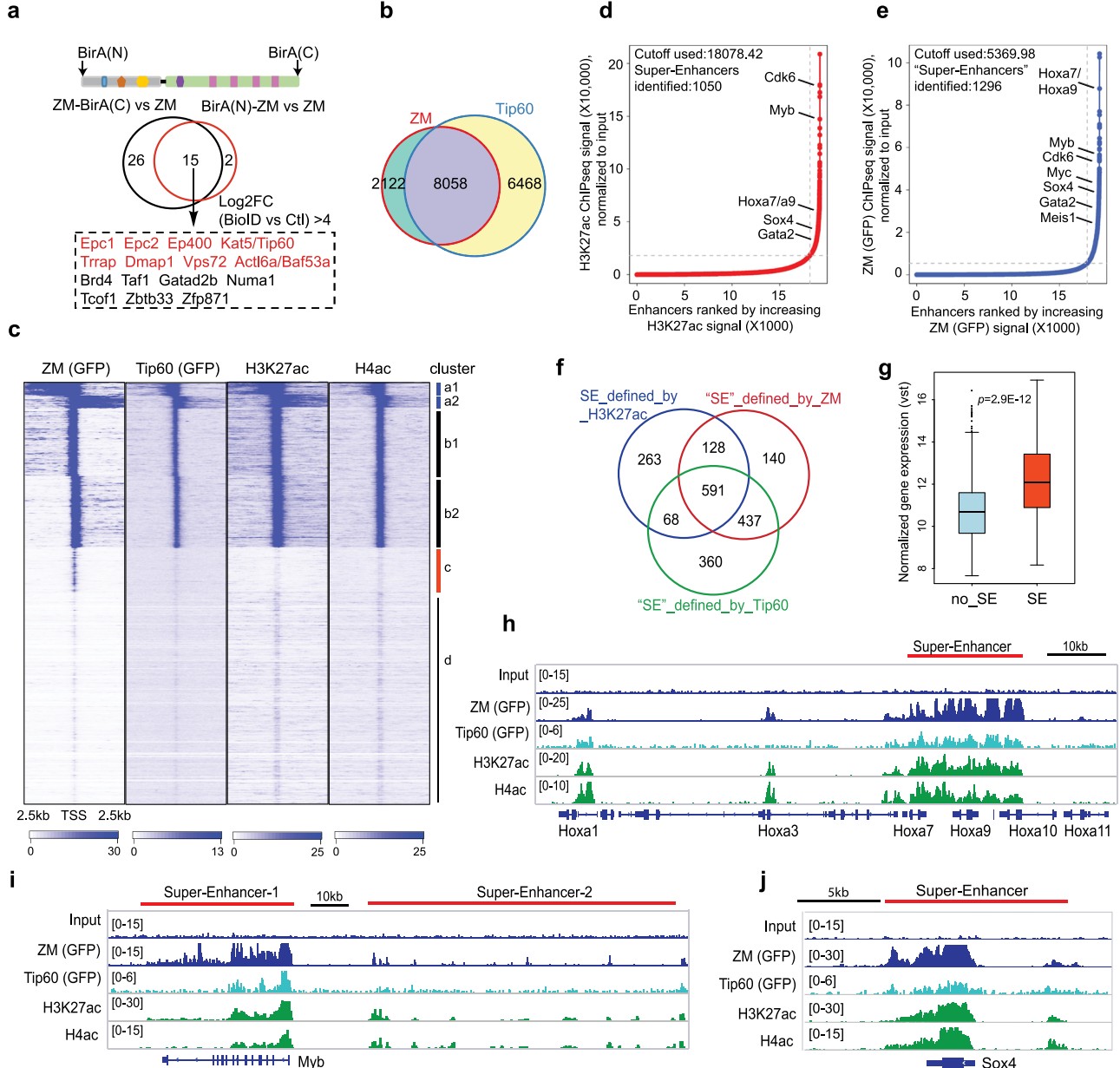

**Fig. 4 ZM interacts with the NuA4/Tip60 complex, generating super-enhancers characterized by dense histone acetylation, typically seen at proto-oncogenes. a** Venn diagram shows the ZM-interacting proteins (bottom panel, within dashed frame) identified from two independent experiments of BioID followed by mass spectrometry using murine HSPCs transduced with ZM tagged by a BirA ligase at either its N-terminus or C-terminus (top panel). HSPCs immortalized by ZM without a BirA tag were used as negative control (with a cutoff of log2[fold-change]>4). **b** Venn diagram displaying overlap between ZM and Tip60 ChIP-seq peaks. **c** Clustered heatmaps showing co-localization of ZM, Tip60, H3K27ac and H4ac at promoter-proximal regions (+/−2.5 kb from TSS) in ZM-transformed AML cells. Four clusters (labeled as a to d, same as Fig. 3e) were produced based on their distinct ZM ChIP-seq signals. TSSs on the two DNA strands are labeled as 1 and 2 (such as a1/2 and b1/2), respectively. **d, e** Hockey-stick plot shows distribution of input-normalized ChIP-seq signals of H3K27ac (**d**) or ZM (**e**) across all enhancers annotated by H3K27ac peaks (promoter-proximal regions or TSS +/−2.5 kb were excluded). Representative proto-oncogenes associated with super-enhancers (SEs), called by ROSE[83,84], are indicated. **f** Venn diagram illustrates overlaps among SEs called based on H3K27ac, ZM or Tip60 ChIP-seq signals. **g** Boxplot showing overall expression levels of genes directly upregulated by ZM, either SE-associated (n = 101) or without SE (no_SE, n = 895). The p value was calculated by two-sided Student's t test. The line in the middle of the box marks the median. The vertical size of box denotes the interquartile range (IQR). The upper and lower hinges correspond to the 25th and 75th percentiles. The upper and lower whiskers extend to the maximum and minimum values that are within 1.5 × IQR from the hinges. Outliers beyond the whiskers are plotted as circles. **h–j** IGV view of normalized ChIP-seq signals at the indicated gene. SEs defined by H3K27ac are depicted with a red bar. The read counts were first normalized to RPGC and then normalized to input.

the NuA4/Tip60 complex is required for ZM-induced transformation, we introduced into ZM the damaging mutations that we have shown by coIP to abolish ZM:Tip60 interaction (Fig. 5a, b), which include a deletion of MBT repeats (ΔMBTx4) and

ALL > DDD, triple point mutations (A661D/L684D/L688D) of MBTD1's MBT1-2 motifs recently shown to disrupt hydrophobic binding of WT MBTD1 to EPC1, a subunit of NuA4/Tip60 complex[22]. Post-transduction into murine HSPCs (Fig. 5c), both

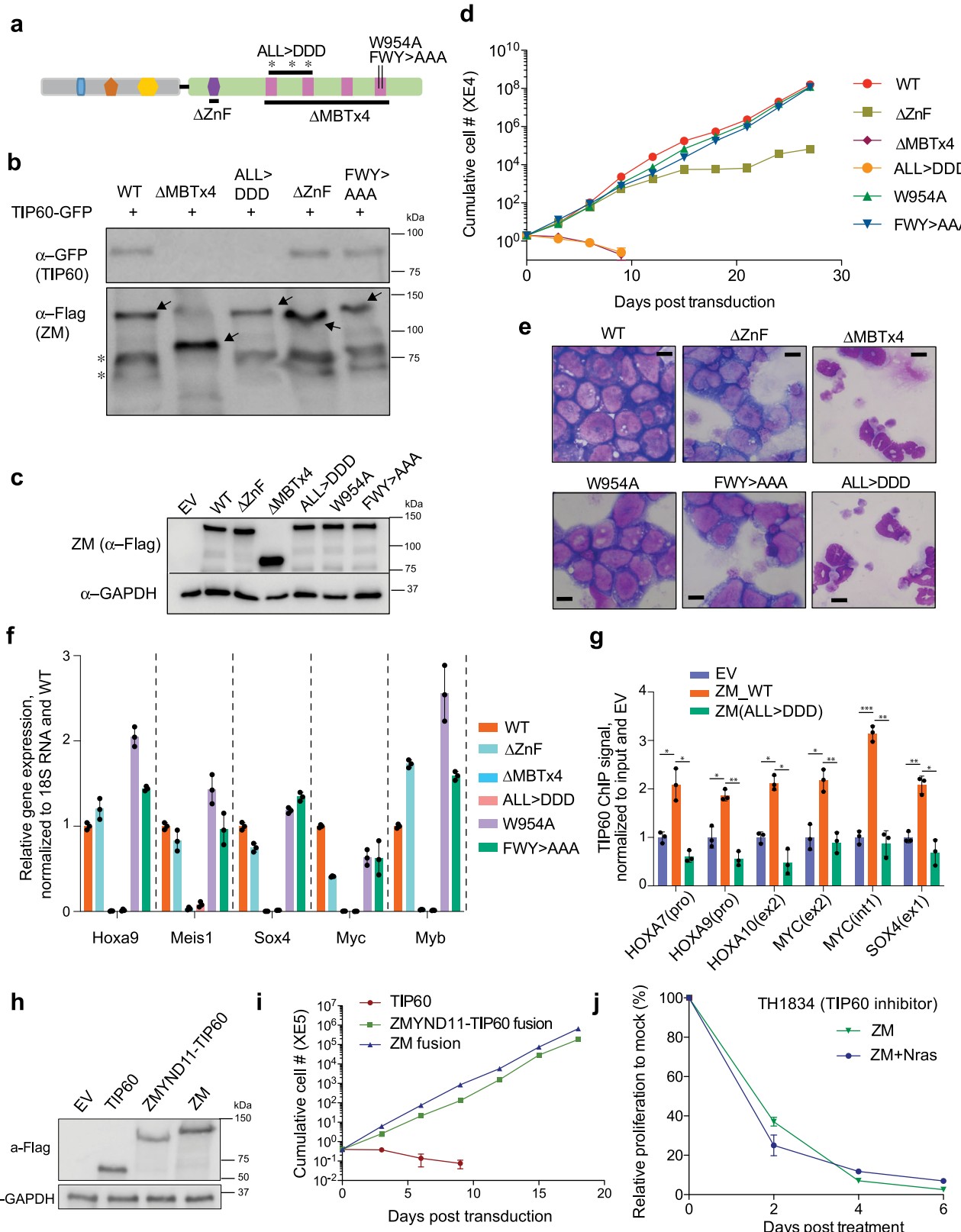

Tip60-interaction-defective mutants failed to induce immortalization of HSPCs, compared to WT (Fig. 5d). In addition to interacting with EPC1/Tip60 complex, MBTD1 harbors an uncharacterized zinc finger (ZnF) domain and a H4K20me1/2-recognizing motif within its fourth MBD repeat (MBT4)[20]. Similar to WT, the ZnF-deleted (ΔZnF) and H4K20me1/2-

binding-defective mutants (Fig. 5a; W954A and F951A/W954A/Y958A [FWY > AAA]) all interacted with Tip60 efficiently in the coIP assay (Fig. 5b) and all induced transformation of HSPCs (Fig. 5c, d), albeit HSPCs transduced with ZnF-deleted ZM showed a mildly reduced proliferation after long-term cultivation. In contrast to WT ZM and its H4K20me1/2-binding-defective or

**Fig. 5 Recruitment of the NuA4/Tip60 complex is essential for ZM-induced target gene activation and oncogenic transformation. a** Scheme showing the used mutations at ZM's MBDT1 segment. ALL > DDD and FWY > AAA indicate mutations of A661D/L684D/L688D and F951A/W954A/Y958A, respectively. **b** CoIP for assessing interaction between the indicated 3XHA-3XFlag-tagged ZM and GFP-tagged Tip60 in 293 T cells. **c** Immunoblotting for ZM (3XHA-3XFlag-tagged) stably expressed in HSPCs. **d** Proliferation kinetics of murine HSPCs post-transduction of ZM, either WT or the indicated mutant. $n = 3$ biological replicates per group and data is presented as mean ± SD. **e, f** Wright-Giemsa staining (**e**; scale bar = 10 um) and RT-PCR of the indicated oncogenic TFs (**f**) using murine HSPCs ten days post-transduction of ZM, either WT or the indicated mutant. qPCR signals from three independent experiments were normalized to 18S RNA and then to WT and presented as mean ± SD. **g** ChIP-qPCR quantifying binding of endogenous TIP60 at the indicated gene in 293T cells stably expressing empty vector (EV) or ZM, either WT or a TIP60-binding-defective mutant (ALL > DDD). ChIP signals from three independent experiments were normalized to input and then to EV and presented as mean ± SD. The $p$ values were calculated by two-sided Student's t test and denoted as follows: *$p < 0.05$; **$p < 0.01$; ***$p < 0.001$. The exact $p$ values (from left to right) are: 0.039, 0.016, 0.039, 0.007, 0.018, 0.02, 0.016, 0.007, 3.89E-5, 0.006, 0.008, 0.03. **h** Immunoblotting showing stable expression of the indicated 3xHA-3xFlag-tagged protein in murine HSPCs. **i** Proliferation of HSPCs post-transduction of TIP60, ZMYND11-TIP60 fusion, or ZM ($n = 3$ biological replicates per group, with data presented as mean ± SD). **j** Growth of ZM and ZM + Nras$^{G12D}$ AML cells post-treatment with 25uM TH1834, after normalization to mock-treated controls ($n = 3$ biological replicates per group, with data presented as mean ± SD).

ZnF-deleted mutants, the Tip60-interaction-defective mutants (ΔMBTx4 or ALL > DDD) failed to arrest terminal differentiation (Fig. 5e) and failed to maintain high expression of ZM's pro-oncogenic targets such as Hoxa9, Meis1, Sox4, Myc and Myb (Fig. 5f).

To further test the hypothesis that ZM recruits NuA4/Tip60 complex to target loci for gene activation, we generated HEK293T cells stably expressing WT ZM or ZM$^{ALL>DDD}$, and subsequent ChIP-qPCR for endogenous TIP60 showed that only WT ZM, not ZM$^{ALL>DDD}$, was able to enhance TIP60 recruitment to proto-oncogenes compared to EV control (Fig. 5g). Additionally, we have swapped ZM's MBTD1 segment with TIP60 (Supplementary Fig. 6a) and found that, just like ZM, such an artificial fusion of ZMYND11-TIP60 conferred primary HSPCs indefinite self-renewal, an activity not seen with TIP60 alone (Fig. 5h, i), which strongly argues that ZM's MBTD1 fusion segment exerts its oncogenic function primarily through recruiting NuA4/TIP60 complex. Consistently, ZM-transformed AML cells showed a considerable sensitivity to treatment of TH1834 (Fig. 5j), a selective enzymatic inhibitor of TIP60[37], which reversed ZM-mediated proto-oncogene activation (Supplementary Fig. 6b).

The above observations thus demonstrate that recruitment of NuA4/TIP60 complex by ZM to target genes is essential for ZM-mediated oncogenesis and trans-activation of downstream proto-oncogenes.

**An H3.3K36me3-binding PWWP domain harbored within ZM is essential for sustaining the chromatin association and leukemogenesis of ZM.** Next, we sought to understand how fusing MBTD1 with ZMYND11's N-terminal segment converts it to an oncoprotein. Given that the tandem Bromo-PWWP domain of ZMYND11, a module retained within ZM (Fig. 1a), was previously shown to be a 'reader' specific for H3.3K36me3[18,19,38], we reasoned that the ZMYND11 fusion segment contributes to genomic targeting of ZM. Towards this end, we first performed mutagenesis of ZMYND11 motifs to score their requirement for ZM-mediated transformation (Fig. 6a). Deletion of the PHD motif (ΔPHD), or introduction of a point mutation at PWWP residues known to be essential for ZMYND11's H3.3K36me3 binding (W294A and F310A; Fig. 6a)[18], abrogated ZM-mediated immortalization of HSPCs (Fig. 6b). Consistently, transduction of HSPCs with the PHD-deleted or PWWP-mutated ZM, unlike WT control, failed to block terminal myeloid differentiation (Fig. 6c) and failed to maintain activation of proto-oncogenes (Hoxa, Meis1, Sox4, Myc and Myb; Fig. 6d). To further test the hypothesis that the H3.3K36me3-'reading' PWWP domain contributes to genomic targeting and thus pathogenesis of ZM, we first verified that only WT and not the PWWP-mutated ZM efficiently associates with H3.3K36me3, not H3.3K36me2, in a

peptide pulldown assay (Fig. 6e and Supplementary Fig. 7a). Furthermore, although WT and PWWP mutants of ZM displayed a similar exclusive nuclear localization in 293T stable expression cells (Supplementary Figs. 4a and 7b), a subsequent chromatin fractionation assay demonstrated that their distribution patterns were different– a significantly more fraction of PWWP-mutated ZM were dissociated from chromatin to nucleoplasm, relative to WT or PHD-deleted controls (Fig. 6f); additionally, global removal of H3K36me3 by knocking down SETD2 also led to reduction in overall chromatin binding of WT ZM (Fig. 6g). These observations lent strong support for a role of H3K36me3-'reading' capability harbored within PWWP in mediating and/or stabilizing ZM's chromatin occupancy. Indeed, ChIP-qPCR verified that, compared to WT ZM, its PWWP mutants displayed a dramatically reduced binding to pro-leukemic genes that we have shown to be co-bound by ZM and H3K36me3 in AML cells (Fig. 6h). Together, these results demonstrate that an H3.3K36me3-engaging PWWP motif in ZMYND11 part is critical for ZM's chromatin association, as well as ZM-mediated proto-oncogene activation and oncogenic transformation.

**Genomic binding locations of ZM are determined by both ZMYND11 and MBTD1 segments.** Interestingly, although ZM relies on its H3K36me3-'reading' PWWP domain for a more stable association with chromatin (Fig. 6f), its genomic distributions do not fully overlap with H3K36me3 (Fig. 3e, f). Considering that ZM interacts with the NuA4/TIP60 complex, we speculated that the chromatin targeting and binding pattern of ZM are influenced by both fusion segments. To test this idea, we employed CUT&RUN[39] to profile genome-wide binding of 3XHA-3XFlag-tagged ZM, either full-length or with either fusion segment deleted (ΔM or ΔZ; Fig. 1a, b), post-transduction into 293T cells. Replicated CUT&RUN assays using either Flag or HA-specific antibody identified highly similar ZM binding sites (Supplementary Fig. 7c). Similar to what was observed in AML cells, full-length ZM mainly binds to TSS-proximal regions and gene bodies in 293T cells (Fig. 6i, j). Notably, the MBTD1 part deletion in ZM significantly shifted its binding towards gene body, suggesting that the MBTD1 segment confers ZM a TSS-targeting capability (Fig. 6i, j and Supplementary Fig. 7d, e). In contrast, deletion of the ZMYND11 segment resulted in relocation of the fusion almost exclusively to TSS, indicating an expected gene body-targeting role of this segment (Fig. 6i, j and Supplementary Fig. 7d, e). Intriguingly, although ZMΔM is more enriched in the gene body compared to full-length ZM, a subset of its peaks was also found at TSSs, implying that the ZMYND11 part harbors an unknown TSS-binding activity (Fig. 6i, j and Supplementary Fig. 7d, e). Pearson correlation analysis revealed a closer similarity between the binding pattern of full-length ZM

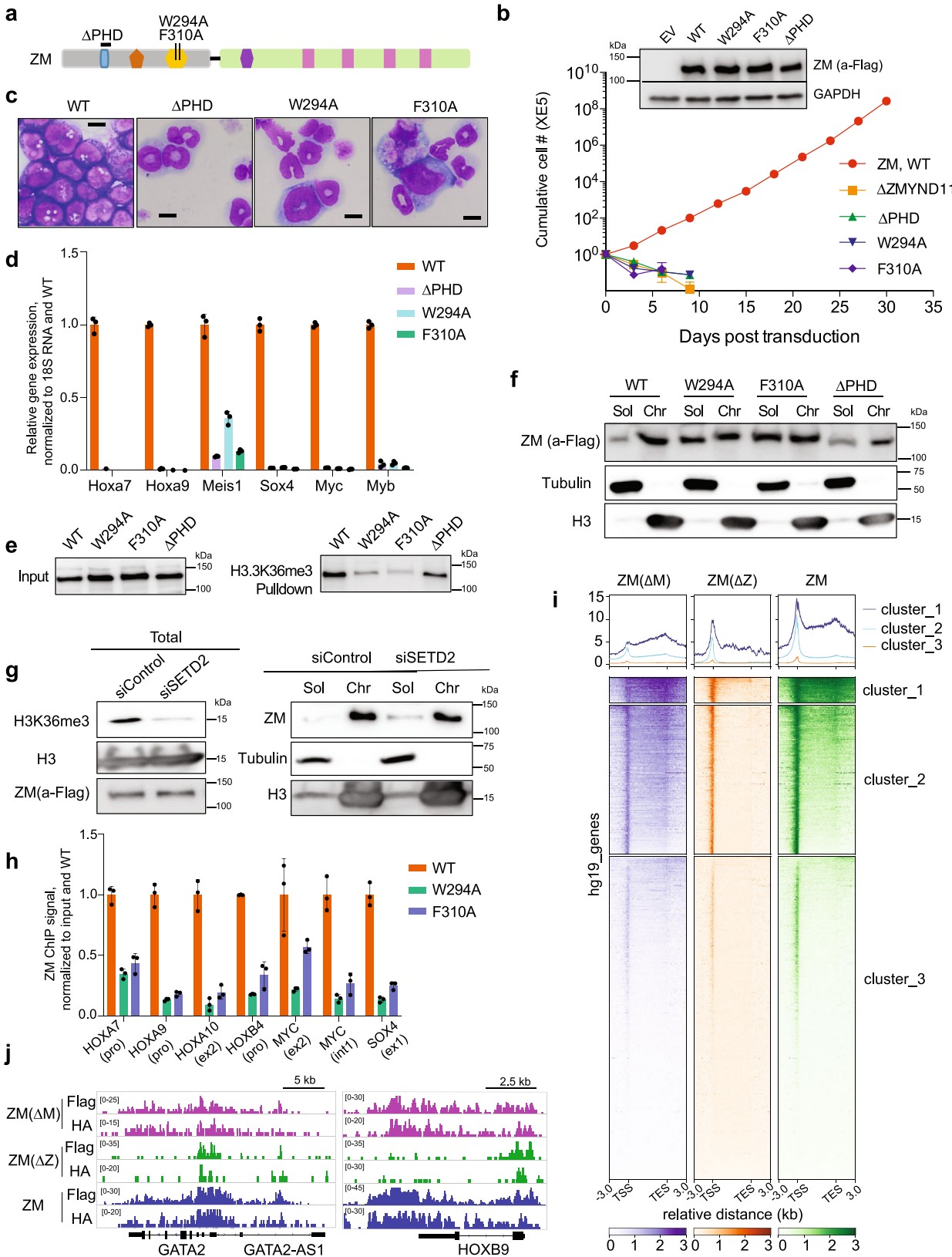

and that of ZMΔM, relative to ZMΔZ, suggesting that the ZMYND11 segment plays a major role in ZM's chromatin targeting (Supplementary Fig. 7c). Together, chromatin targeting of ZM is modulated by both ZMYND11 segment (for binding location and affinity) and MBTD1 segment (for binding location), with the former one being the major determinant.

**Brd4 inhibition suppresses ZM-mediated aberrant stemness gene activation and leukemic transformation**. In our effort to define ZM's protein interactome via BioID, we also identified Brd4 among the top hits (Fig. 4a and Supplementary Fig. 5b). This finding is in agreement with reports that Brd4 functions to 'read' histone acetylation with a preferred binding to multi-acetylated

**Fig. 6 An H3.3K36me3-binding PWWP module is essential for chromatin association of ZM, as well as ZM-enforced proto-oncogene activation and oncogenic transformation. a** Scheme of the used ZM mutations within its ZMYND11 segment. **b** Proliferation of murine HSPCs stably transduced with ZM, WT or mutant (insert, immunoblot of ZM; $n = 3$ biological replicates per group, with data presented as mean ± SD). **c** Wright-Giemsa staining of HSPCs ten days post-transduction of ZM, WT or mutant (scale bar = 10 µm). **d** RT-qPCR of the indicated ZM direct target genes in HSPCs stably transduced with ZM, WT or the indicated mutant. qPCR signals from three independent experiments were normalized to those of 18S RNA and then to WT and presented as mean ± SD. **e** Pulldown using H3.3K36me3-containing peptide and nuclear extract of 293T cells stably expressing 3XHA-3XFlag-tagged ZM, either WT or the indicated mutant. **f, g** Immunoblot for 3XHA-3XFlag-tagged ZM in soluble nucleoplasmic (Sol) and chromatin-bound (Chr) fractions, prepared from 293T cells with stable expression of WT or mutant ZM (**f**) or those WT ZM stable expression cells with SETD2 knock down (siSETD2; **g**) or mock treatment (siControl). SETD2 knockdown led to global reduction of H3K36me3. Tubulin and H3 serve as fractionation controls. **h** ChIP-qPCR examining ZM occupancy to the indicated gene in 293T stable expression cells. ChIP signals from three independent experiments were normalized to those of input and then to WT and presented as mean ± SD. **i** Average signal profiles and clustered heatmaps displaying 3XHA-3XFlag-tagged ZMΔM, ZMΔZ, and ZM CUT&RUN signals (detected with HA antibody) over all genes in 293T cells. **j** IGV views of ZMΔM, ZMΔZ, and ZM CUT&RUN signals (detected with Flag or HA antibody) at the indicated gene in 293T cells.

H4 tails modified by NuA4/TIP60 complex[40]. Using ChIP-seq, we found that ZM and Brd4 displayed a global co-occupancy pattern in ZM-transformed AML cells (Fig. 7a, b; Supplementary Fig. 8a). These findings suggest that Brd4 acts downstream of ZM:Tip60 complex and their catalyzed histone acetylation to stimulate gene transcription, implicating a potential therapeutic avenue due to a 'druggable' nature of Brd4. Thus, we treated ZM-transformed AML cell lines with I-BET151, a selective Brd4/BET inhibitor[41], and found that, I-BET151 treatment efficiently blocked AML cell proliferation and elevated cell death (Fig. 7c). Moreover, Brd4/BET inhibition triggered terminal differentiation of AML blasts (Fig. 7d), inducing expression of myeloid differentiation marker while decreasing that of leukemia stem cell marker (Supplementary Fig. 8b). Mechanistically, these anti-leukemia effects of I-BET151 are likely due to its suppression of ZM-activated pro-leukemic genes, including Hoxa7, Hoxa9, Meis1, Myc and Myb (Fig. 7e). We also treated mice carrying ZM + Nras$^{G12D}$-induced AML with I-BET151. Bioluminescent imaging of live animals showed that I-BET151 administration led to a marked delay of the in vivo leukemic expansion (Fig. 7f). At the experimental end-point when the vehicle-treated cohort showed severe AML symptoms, all I-BET151-treated mice remained healthy, with much less enlarged spleen and lower counts of peripheral white blood cells compared to mock (Fig. 7g and Supplementary Fig. 8c), and survived for a significantly longer period (Fig. 7h). These data collectively support that Brd4 is essential for ZM-driven leukemia development, a process suppressed by bromodomain inhibitor.

## Discussion

In this study, we show that introduction of ZM into primary HSPCs is sufficient to cause transformation in vitro and AML development in animal models, thereby demonstrating that this disease-associated chimeric gene is a bona fide oncogene. Additionally, we have investigated into the molecular mechanisms underlying ZM-mediated leukemogenesis. Integrated RNA-seq and ChIP-seq analyses demonstrated that ZM chimera directly binds to and maintains high transcription of 'stemness'-related genes, including a set of pro-leukemic genes such as Hoxa, Meis1, Myc, Myb and Sox4. Furthermore, our systematic mutagenesis analysis of ZM provided important mechanistic insights—interaction with the NuA4/TIP60 complex (via ZM's MBTD1 segment) and an H3K36me3-'reading' Bromo-PWWP module retained within its ZMYND11 segment are both essential for ZM-mediated transformation of HSPCs and activation of 'stemness'-related gene-expression program. It is noteworthy that these two molecular events act in concert for inducing oncogenesis because, in contrast to ZM or a similarly-transforming artificial fusion of ZMYND11-TIP60, neither full-length MBTD1 (Fig. 1a–c) nor

wild-type TIP60 (Fig. 5h, i) was able to induce transformation under the same settings.

Among the ZM direct target gene signature, we observed a number of 'stemness' genes (such as master TFs and/or proto-oncogenes), which were previously shown to be dynamically regulated and subject to epigenetic silencing during normal hematopoiesis and cell differentiation, notably due to Polycomb Repressive Complex 2 (PRC2)-mediated H3K27me3[42] (Fig. 7i, left). In the case of ZM chimera, it not only induces histone acetylation depositions at cis-regulatory elements (promoters and super-enhancers) of these 'stemness' genes but, importantly, 'senses' H3K36me3, a transcription-associated histone PTM[43] that 'marks' a transiently active state seen at 'stemness' genes during self-renewal and differentiation of HSPCs (Fig. 7i, right). In a sense, ZM chimera can selectively 'target' HSPC sub-populations displaying high expression of 'stemness' genes for transformation by sustaining a continued activation of this pro-gram via recruitment of NuA4/TIP60 complex and/or acetylation of histones. Indeed, H3K27ac antagonizes the PRC2-mediated methylation of the same histone PTM site (H3K27me3). On the other hand, Tip60-catalyzed acetylations such as multi-acetylated H4 tails provide a platform for tethering Brd4-PTEFb complex[44], which in turn boosts the release of Pol-II into productive elongation phase, a process accompanied by accumulation of SETD2-mediated H3K36me3 during new transcription cycles; of note, such SETD2-generated H3K36me3, associated with the newly synthesized transcripts, can be further 'sensed' by ZM, thereby establishing a feed-forward loop; moreover, the PRC2 activity is also suppressed by the active-transcription-associated H3K36me3 in cis[45]. Together, ZM and recruited cofactors entrap and sustain a transcriptionally active chromatin state (with histone acetylations plus H3K36me3) that is refractory to invasion of PRC2 and H3K27me3. This model (see Fig. 7i) is strongly supported by a lack of ZM-mediated transformation due to the damaging mutation at either its TIP60-interacting or PWWP-H3K36me3 binding interface, an equivalent transformation phenotype observed with the artificial chimera of ZMYND11-TIP60, and a hyper-sensitivity of ZM+ AMLs to I-BET151, a bromodomain inhibitor.

Intriguingly, besides AML-related ZM fusion, aberrant chro-mosomal rearrangements detected in endometrial stromal sar-coma patients recurrently target components of NuA4/TIP60 complex such as EP400, EPC1, EPC2, MEAF6 and MBTD1[46–50]. In addition, other histone acetyltransferase classes, including CBP and EP300, were previously reported to be involved in chromo-somal rearrangements in leukemia patients[51,52], driving disease formation. Our findings, together with others, thus highlight a broad role of hijacking a histone acetyltransferase (HAT) activity for oncogenesis, which justifies development of selective HAT inhibitors for anti-cancer therapy. We have also demonstrated

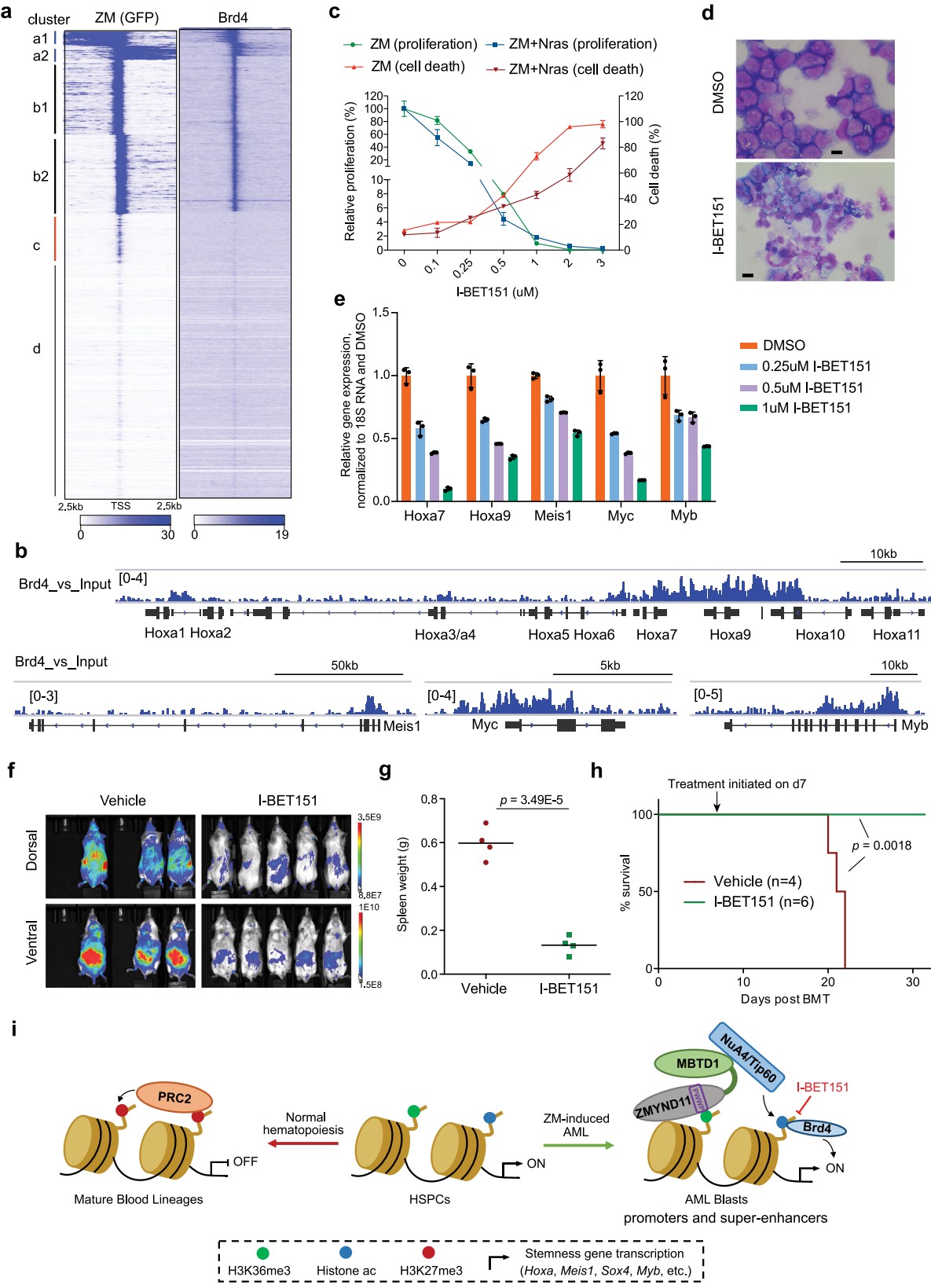

BET inhibitor as an effective treatment for murine ZM+ AMLs. But it is noteworthy to mention that the currently available BET inhibitors as single agents, despite their promising anti-cancer activity in animal models, seem to only exhibit limited efficacy in clinical trials[53]. To address potential issues of low potency, next-generation BET-inhibiting compounds are being developed, such as bivalent BET inhibitors that bind to tandem bromodomains simultaneously and PROTAC-based BET degraders, which offer superior therapeutic effects in preclinical studies[54–57]. In addition, combination therapy with other synergistic anti-cancer drugs was reported to greatly boost the efficacy of BET inhibition and prevent emergency of drug resistance[58,59]. Overall, improved

**Fig. 7 ZM-induced AML is sensitive to Brd4 blockade. a** Heatmaps showing un-supervised k-means clustering of ZM and Brd4 ChIP-seq peaks at all promoter-proximal regions (TSS +/−2.5 kb). Four clusters (labeled as a to d, same as Fig. 3e) were produced based on their distinct ZM ChIP-seq signals. TSSs on the two DNA strands are labeled as 1 and 2 (such as a1/2 and b1/2), respectively. **b** IGV views of normalized Brd4 ChIP-seq signals at indicated loci in ZM-transformed AML cells. The read counts were first normalized to RPGC and then normalized to input. **c** Proliferation (left y-axis, normalized to DMSO-treated cells) and cell death (right y-axis, measured by trypan blue staining) of AML cells transformed by ZM alone or ZM plus $Nras^{G12D}$, post-treatment with the indicated concentration of I-BET151 (x-axis) for four days. $n = 3$ biological replicates per group and data is presented as mean ± SD. **d** Wright–Giemsa staining of ZM-transformed AML cells post-treatment with DMSO or I-BET151 (0.25 uM) for four days (scale bar = 10 μm). **e** RT-qPCR of the indicated gene in ZM-transformed AML cells post-treatment with DMSO or the indicated concentration of I-BET151 for four days. qPCR signals from three independent experiments were normalized to those of 18S RNA and then to DMSO-treated control and presented as mean ± SD. **f** Bioluminescent imaging of mice transplanted with ZM + $Nras^{G12D}$-transformed AML cells, two weeks post-treatment with either vehicle or I-BET151 (30 mg/kg daily IP injection). **g** Weight of spleens in the indicated cohorts ($n = 4$ mice per group) at the study endpoint. The p value was calculated by two-sided Student's t test. **h** Kaplan–Meier survival curve of mock- and I-BET151-treated mice bearing the ZM + $Nras^{G12D}$ AML (n = cohort size). Black arrow indicates the day 7 when compound administration was initiated. The p value was calculated by two-sided log-rank test. **i** Model illustrates the ZM-mediated proto-oncogene activation and leukemogenesis.

means of BET inhibition likely aid in more effective anti-cancer therapy.

Furthermore, we found ZM alone induces AML after a long latency with incomplete penetrance, implying that additional cooperating mutations are likely required to drive full-blown disease. Indeed, our in vivo assay using co-expressed $Nras^{G12D}$ and the WES-based identification of a spontaneous activating mutation in Ptpn11 acquired by ZM alone-induced murine AML highlight the activated Ras signaling pathway as a cooperating oncogenic event for ZM. Additionally, given that wildtype ZMYND11 functions as a putative tumor suppressor[16,18], it is likely that disruption of one normal ZMYND11 allele in patients as a consequence of aberrant chromosomal translocation might contribute to malignant transformation as well. Finally, in contrast with what we observed with the PWWP mutations, deletion of the PHD domain within ZM only mildly impaired H3K36me3 binding in vitro, yet it similarly abolished leukemogenic potentials of ZM, pointing to a possible H3K36me3 binding-independent oncogenic function of this domain. These unsolved questions warrant additional investigation in future.

## Methods

**Plasmid construction.** The human ZMYND11-MBTD1 (ZM) fusion cDNA was generated by Gibson Assembly-mediated ligation between a ZMYND11 cDNA segment encoding its amino acids 1-409 (NM_006624) and a MBTD1 segment (NM_017643), which contains a small region of 5'-UTR immediately upstream of the start codon (which encodes 16 amino acids; see Fig. 1a) plus the entire open reading frame (ORF, amino acids 1-628)[8]. The ZM chimeric cDNA was then C-terminally tagged by either 3XHA-3XFlag or GFP and cloned into the MSCV-Puro retroviral expression vector (Clontech). For BioID-based protein interactome studies, a biotin ligase (BirA) cDNA (a kind gift of BD Strahl) was inserted into either N-terminus or C-terminus of ZM cDNA in the MSCV-Puro vector. Human TIP60 ORF cDNA (NM_006388.4) was C-terminally tagged by GFP and then cloned into MSCV-Neo vector. An 3XHA-3XFlag-tagged artificial fusion of ZMYND11-TIP60 was generated by swapping the MBTD1 segment with the entire human TIP60 cDNA through restriction enzyme digestion and ligation. Internal deletion or point mutation was created by an overlapping PCR protocol[60]. For coexpression of ZM plus $Nras^{G12D}$, the latter cDNA was inserted downstream of a luciferase-IRES (internal ribosome entry site) cassette in a home-made MSCV-Neo-based bicistronic vector. A MSCV-Neo-luciferase construct was a kind gift of Q Zhang (UT Southwestern). All plasmids used were confirmed by sequencing.

**Antibodies.** Antibodies used for western blots (all diluted by 1:2000) include anti-Flag (M2)-HRP (Sigma, A8592), anti-H3 (CST, 9715), anti-GAPDH (CST, 2118), anti-β-Tubulin (CST, 2146), Streptavidin-HRP (CST, 3999) and anti-GFP (CST, 2956). Antibodies used for FACS (all diluted by 1:100) include c-Kit^APC (Invitrogen, 17-1172-82), c-Kit^FITC (eBioscience,11-1171-85), Cd34^APC (eBioscience, 50-0341-82), Cd34^FITC (BD, 560238), Mac1^APC (BD, 557686), Mac1^FITC (eBioscience, 11-0112-85), Gr1^FITC (eBioscience, 11-5931-85), Cd4^FITC (eBioscience,11-0042-82), Cd8a^FITC (eBioscience,11-0081-82), and Cd19^FITC (eBioscience,11-0193-82). Antibodies used in ChIP, ChIP-seq and CUT&RUN assays include anti-Flag (Sigma, F1804), anti-HA (Abcam, ab9110), anti-GFP (Abcam, ab290), anti-H3K36me3 (Abcam, ab9050), anti-H3K27ac (Abcam, ab4729), anti-H3K27me3 (Millipore, 07-449), anti-H4ac (Millipore, 06-866),

anti-BRD4 (Bethyl, A301-985A100) and anti-Tip60 (Santa Cruz, sc-166323). 10 ug antibodies were mixed with 100 μl Dynabeads for each ChIP or ChIP-seq assay. All antibodies used in CUT&RUN assays were diluted by 1:100.

**Purification, expansion, and retroviral transduction of primary murine hematopoietic stem/progenitor cells (HSPCs).** The primary lineage-negative (Lin-) murine HSPCs were obtained through isolating bone marrow cells from femur and tibia of wildtype BALB/C mice, followed by depletion of mature hematopoietic cells using a lineage cell depletion kit (Miltenyi Biotec, 130-090-858) as described before[3,4]. Lin- HSPCs were then subject to cytokine stimulation for ex vivo expansion in the Opti-MEM medium supplemented with 15% of FBS (Invitrogen, 16000-044), 1% of antibiotics, 50 μM of β-mercaptoethanol, 25 ng/mL of murine SCF (PeproTech, 250-03) and 10 ng/ml of murine Flt3 ligand (Sigma, SRP3198). After 3–4 days of stimulation, spinoculation was performed for retro-viral transduction. Briefly, murine HSPCs were mixed with concentrated retrovirus (Retro-X™ Concentrator; Takara, 631456) in the presence of 8 ug/ml polybrene in a fibronectin-coated 6-well plate, followed by centrifugation at 1,500 g for 1 h. Drug selection with 2 μg/mL puromycin or 500 μg/mL G418 started 2 days post-infection. For routine liquid culture of transduced HSPCs, we used a home-made medium recipe, which uses the culture supernatants of an mSCF-producer cell line (mSCF-CHO cells, gift of MP Kamps, UCSD) and an mFlt3l-producer cell line (SP2.0-mFlt3L cells, gift of R. Rottapel, University of Toronto) as source of murine SCF and Flt3L, respectively[5,61,62].

**Sorting and culture of murine HSC and GMP.** Murine bone marrow (BM) cells were isolated from C57BL/6 mice by crushing femurs, tibias, pelvic bones and spine using a mortar and pestle, followed by magnetic enrichment of cKit+ BM cells using the autoMACS machine. cKit+ BM cells were then stained with a combination of fluorochrome-conjugated antibodies and the viability stain propidium iodide (PI), followed by sorting for HSC (Lin-/cKit+/Sca1+/Cd16/32-/Cd150+/Cd48−) and GMP (Lin-/cKit+/Sca1-/Cd16/32+/Cd150−). Purity of sorted cells was found to be ≥ 95%. HSC cells were pre-stimulated in base medium (Opti-MEM, 15% FBS, 1% PS, 50uM β-mercaptoethanol) supplemented with 20 ng/ml mSCF and 100 ng/ml mTPO for 24 h before infection. GMP cells were directly infected without pre-stimulation in the same base medium supplemented with 20 ng/ml mSCF and 10 ng/ml mIL3 and mIL6.

**Phenotypic assays of hematological cells, such as colony-forming unit (CFU), Wright-Giemsa (W&G) staining and proliferation assay.** CFU assay was done using the methylcellulose-based semi-solid medium (STEMCELL Technologies, M3434) according to the manufacturer's protocol. In brief, 8000 freshly infected and drug-selected HSPCs were added to the complete MethoCult medium at a 1:10 (v/v) ratio, then dispensed into a non-treated 35-mm dish using blunt-end needle attached to a sterile syringe, and incubated for seven days before counting. CFU number was manually counted using 60-mm gridded scoring dish and inverted microscope. After counting, cells from colonies were harvested for replating by diluting the semi-solid medium with PBS and centrifugation. For W&G staining, the cytospin coverslip containing a monolayer of 100,000 cells was sequentially stained with 100% of Wright stain solution (Sigma, WS16) for two minutes and 10% Giemsa stain solution (Sigma, GS500) for seven minutes prior to mounting and imaging with the EVOS XL Core Imaging System (Invitrogen). Counting-based proliferation assays of HSPCs or AML cells were carried out as previously described[63].

**Immunofluorescence staining and flow cytometry (FACS) analysis.** Cells were washed and suspended in FACS stain buffer (5% FBS in PBS) to a final density of 10 million cells per mL, followed by incubation with fluorescently (either FITC, APC or PE) conjugated primary antibodies at 1:100 dilution for 30 min on ice,

washing with FACS stain buffer, and analysis using a Thermo Fisher Attune NxT machine (available from the UNC Flow Cytometry Core Facility). Single-stained cells were prepared as compensation controls while the unstained cells were used to set negative gate. Data were analyzed using FlowJo software.

**Murine bone marrow transplantation (BMT)-based leukemogenic assay**. All animal experiments were approved by and performed in accord with the guidelines of the Institutional Animal Care and Use Committee at the University of North Carolina (UNC). BALB/C mice were purchased from the Jackson Laboratory and maintained by the Animal Studies Core affiliated to the UNC Lineberger Comprehensive Cancer Center. Mice were housed in a germ-free environment with food and tap water ad libitum. RT and relative humidity were held at 22 °C ± 2 °C and 30–70%, respectively. Automatic light control guaranteed a 12-h light/dark cycle (7:00 to 19:00/19:00 to 7:00). BMT and in vivo leukemogenic assays were carried out by tail-vein injection of 1 million transduced HSPCs into sub-lethally irradiated (300 rads), syngeneic Balb/c mice (performed by UNC Animal Studies Core) as previously described[4,5]. Transplanted mice exhibiting clinical signs of leukemia, such as hunched posture, hind limb paralysis, poor mobility, labored breathing and splenomegaly, were sacrificed, followed by isolation and pathological analysis of blasts-infiltrating tissues, such as femur bone and enlarged spleen, by Hematoxylin and eosin (H&E) staining (carried out by UNC Histology Research Core). Primary leukemic blasts were also purified and introduced into secondary recipient mice. At termination, blood collected from the posterior vena cava was subject to complete blood count (CBC) by a Hemavet 950FS machine (Drew Scientific).

**Chromatin fractionation**. Whole cell lysates were fractioned into soluble (including cytoplasm and nucleoplasm) and chromatin-bound fractions as previously described[64] with the following modifications. In brief, cells were lysed on ice for 20 min in cold buffer containing 10 mM PIPES (pH 7.0), 300 mM sucrose, 200 mM NaCl, 3 mM MgCl$_2$, 0.5% Triton X-100 and 1× EDTA-free protease inhibitor (Roche), followed by centrifugation at 1300 g for 5 min at 4 °C to separate the supernatant with the pellet, which represents soluble and chromatin-bound fractions, respectively. Lysates containing equal numbers of cells were loaded on SDS-PAGE for western blot analysis. Histone H3 and β-tubulin antibody was used as immunoblotting controls for monitoring the purity of fractionation.

**Co-immunoprecipitation (CoIP)**. HEK293T cells were harvested from one confluent 10 cm dish 48 h post-transfection and lysed for 1 h at 4 °C with rotation in 1 mL IP350 buffer (20 mM Hepes pH 7.9, 350 mM NaCl, 0.2% NP40, 0.4 mM EDTA, 1.5 mM MgCl$_2$, 10% glycerol) freshly added with 1 mM DTT, 1X protease inhibitor cocktail, 1 mM PMSF, and 250 units of Benzonase (Sigma, E1014), followed by addition of equal volume of IP0 buffer (no salt and detergent, with everything else same as IP350 buffer) to bring down the salt and detergent concentration (IP175 buffer: 175 mM NaCl and 0.1% NP40). The cleared cell lysates were incubated with anti-FLAG M2 magnetic beads (Sigma, M8823) overnight at 4 °C, followed by intensive washing with the IP175 buffer. The bound proteins were subject to western blotting analysis.

**Peptide pull-down assay**. Peptide pull-down assay was performed as described previously[4,65] with slight modifications. Briefly, biotinylated histone peptide, which contains histone H3.3 amino acids 27–46 with di- or tri-methylated Lys-36 (i.e. H3.3K36me2/3), was incubated with NeutrAvidin agarose (Thermo Fisher, 29204) in PBS buffer for 4 h. The peptide-resin was then washed three times with 1 mL of washing buffer (PBS plus 0.1% TritonX-100) to remove unbound peptide before overnight incubation at 4 °C with the whole cell lysate prepared from 293T cells stably expressing ZM (lysate in a final IP175 buffer as described above). After binding, the resin was washed extensively in IP175 buffer for five times and analyzed by western blot.

**RNA sequencing (RNA-seq)**. RNA-seq was performed as described before[66,67]. In brief, total RNAs were extracted using RNeasy Plus kit (Qiagen, 74136) and residual DNAs were removed using Turbo DNA-free kit (Thermo Fisher, AM1907) to ensure purity of RNA sample. RNA-seq libraries were generated using NEBNext Poly(A) mRNA Magnetic Isolation Module (NEB, E7490) and NEBNext Ultra II RNA library Prep kit (NEB, E7770) according to manufacturer's instructions, followed by quality check with Agilent TapeStation system and deep sequencing on the Illumina Nextseq 500 platform (available from UNC cores) using Nextseq 500/550 High Output Kit v2.5 (Illumina, 20024906).

**RNA-seq data analysis**. The fastq files were aligned to the mm10 mouse genome (GRCm38.p4) using STAR v2.4.2 with the following parameters: –outSAMtype BAM Unsorted –quantMode TranscriptomeSAM[68]. Transcript abundance for each sample was estimated with salmon v0.1.19 to quantify the transcriptome defined by Gencode gene annotation[69]. Gene level counts were summed across isoforms and genes with low expression (all samples had less than 10 reads) were removed before downstream analyses. DESeq2 was used to test for differentially expressed genes (DEGs) between different samples as previously described[66,70]. Gene expression

heatmaps were generated using mean-centered log2 converted TPM (Transcripts Per Million) sorted in descending order based on their expression values in ZM-immortalized AML cells and R's package "gplots" v3.0.3 with either no clustering or column hierarchical clustering by average linkage. Volcano plots visualizing the DEGs were produced using R's package "EnhancedVolcano" v 3.11 with the indicated filters. Functional annotation of DEGs was done using web-based software: The Database for Annotation, Visualization and Integrated Discovery (DAVID) v6.8[71].

**Identification of murine Lin-Sca1+ c-Kit+ (LSK) hematopoietic stem cells-associated stemness gene signature**. DESeq2 was used to identify the differentially expressed genes between primitive LSK cells and each of a set of mature/differentiated hematological cell types (including B cell, CD4T, CD8T, Monocyte, Macrophage, Erythrocyte, and Neutrophil) using the published RNA-seq dataset[72]. Genes that are commonly upregulated in LSK cells, relative to all mature cells, with a cut-off of Log2FC over 1 and adjusted p value less than 0.01, were selected as LSK 'stemness' genes (922 genes in total). The unique enrichment of the defined 922 LSK 'stemness' genes in self-renewing hematopoietic stem cells was verified by Gene Set Enrichment Analysis (GSEA) analysis using different published RNA-seq datasets and customized gene set[73,74].

**Chromatin immunoprecipitation (ChIP) followed by deep sequencing (ChIP-seq)**. Independently derived, ZM-established AML cells were used for ChIP-seq as described before[66]. In brief, cells were fixed in PBS containing 1% formaldehyde (Thermo Scientific, 28908) for 10 min, and then quenched by addition of 125 mM of glycine for 5 min. The fixed cells were first lysed in LB1 buffer (50 mM HEPES-KOH pH 7.5, 140 mM NaCl, 1 mM EDTA, 10% glycerol, 0.5% NP-40, 0.25% TritonX-100) and washed in LB2 buffer (10 mM Tris-HCl pH 8.0, 200 mM NaCl, 1 mM EDTA, 0.5 mM EGTA) to purify nuclei, which were then lysed in LB3 buffer (10 mM Tris-HCl pH 8.0, 100 mM NaCl, 1 mM EDTA, 0.5 mM EGTA, 0.1% Sodium Deoxycholate, 0.5% N lauroylsarcosine) and sonicated with the Bioruptor sonicator (Diagenode, B01020001; at high energy setting for 60 cycles with 30 s on and 30 s off). After sonication, 1% TritonX-100 was added to solubilize the nuclear membrane before centrifugation (20,000 g for 10 min at 4 °C). The supernatant was then incubated with antibody-bound dynabeads (Invitrogen, 11204D) overnight at 4 °C. After sequential washes with the previously described buffers (i.e., low-salt buffer, high-salt buffer, LiCl buffer, and TE buffer), the DNA-protein complexes were eluted and subject to reverse crosslink, RNase (Roche, 11119915001) and Protease K (Roche, 03115828001) digestion, and DNA recovery by Qiagen PCR purification kit (Qiagen, 28106). The ChIP-seq libraries were prepared using NEBNext Ultra II DNA Library Prep Kit (NEB, E7645L) following the manufacture's protocol. ChIP-seq libraries were sequenced on an Illumina Nextseq 500 Sequencer using Nextseq 500/550 High Output Kit v2.5.

**ChIP-seq data analysis**. ChIP-seq reads were aligned to the mouse genome (mm10) using BWA (v0.7.15)[75]. After removing reads with a map score < 20, peaks were called by the software MACS2 (v2.1.1)[76] using input as controls and the option (−q 0.1 −m 20 100). ZM peaks called in both GFP- and HA-tagged ChIP-seq samples were considered as common ZM peaks. Broad peaks for H3K27me3 and H3K36me3 were determined using a sliding window approach to detect regions with >2 higher signals in ChIP than in input sample, as previously described[3]. ChIP-seq peaks were associated to genes (coding and non-coding) in this order— assigned as "promoter proximal" (±2.5 kb of transcription start site (TSS)), "gene body", "promoter distal" (−50 kb to −2.5 kb of TSS and +2.5 kb of TSS to +5 kb of transcription ends, excluding those at "gene body"), and otherwise "intergenic", with a minimum of 1-bp overlap. The ChIP-seq read densities around transcription start sites were calculated and clustered using the program seqMINER (v1.3.4)[77]. To determine the overlap of any two lists of peaks, we first merged the two lists to generate a non-overlap discrete list of peaks and then computed its overlap with each of the two original lists. As such, the reported numbers in the Venn diagrams may not add up to the total number of peaks in individual lists. Deeptools were used to generate the input-normalized ChIP fragment coverage bigwig files and heatmaps showing ChIP signals along the transcription unit with the options (computeMatrix scale-regions -b 3000 -a 3000 –regionBodyLength 5000 –skipZeros –sortUsing median)[78]. We did two-step normalization when generating the bigwig coverage file from BAM alignment file— the number of reads per bin was first normalized to 1x genome coverage (reads per genome coverage, RPGC) and then normalized to input by subtract using deepTools bamCoverage and bigwigCompare functions, respectively. The Genomic Regions Enrichment of Annotations Tool (GREAT)[79] was used to determine enriched gene functions and Integrative Genomics Viewer (IGV) was used to display RPGC and input-normalized ChIP-seq signals (bigwig format).

**CUT&RUN**. Cleavage Under Targets & Release Using Nuclease (CUT&RUN) was performed according to the EpiCypher CUTANA CUT&RUN Protocol. In brief, one million of live cells were harvested, washed and re-suspended in wash buffer (20 mM HEPES, pH 7.5, 150 mM NaCl, 0.5 mM Spermidine and 1x Roche Complete Protease Inhibitor), followed by incubation at RT for 10 min with activated ConA magnetic beads (Bangs Laboratories, cat# BP531) which were washed

and re-suspended in bead activation buffer (20 mM HEPES, pH 7.9, 10 mM KCl, 1 mM $CaCl_2$, 1 mM $MnCl_2$). After binding to activated beads, cells were permeabilized in antibody buffer (wash buffer + 0.01% digitonin + 2 mM EDTA) and incubated with HA or Flag antibody (1:100 dilution) on nutator overnight at 4 °C. On the next day, the cell-bead slurry was washed twice with cold digitonin buffer (wash buffer + 0.01% digitonin) and then incubated with pAG-MNase (1:20 dilution, EpiCypher, cat# 15-1116) for 10 min at RT, followed by two washes with cold digitonin buffer. MNase was then activated by addition of $CaCl_2$ to cleave targeted chromatin for 2 h at 4 °C. After chromatin digestion, MNase activity was stopped and chromatin fragments released into supernatant by adding stop buffer (340 mM NaCl, 20 mM EDTA, 4 mM EGTA, 50 μg/ml RNase A, and 50 μg/ml Glycogen) and incubating at 37 °C for 10 min. The DNA was purified from the collected supernatant using the NEB Monarch DNA Cleanup Kit (NEB, cat# T1030) per manufacturer's instruction. Finally, 5–10 ng of purified CUT&RUN-enriched DNA was used to prepare Illumina library using the NEB Ultra II DNA Library Prep Kit per manufacturer's instructions.

**CUT&RUN data analysis**. Raw sequencing reads were first trimmed by Trim Galore to remove low-quality and adaptor bases, and then aligned to hg19 using STAR. After mapping reads to the genome, only primary alignments were extracted followed by removal of duplicate reads using the Picard MarkDuplicates tool. The bigwig coverage file was generated from BAM alignment file using the deeptools bamCoverage function and the number of reads per bin was normalized to 1x genome coverage (reads per genome coverage, RPGC). The clustered correlation heatmap was made using the deepTools plotCorrelation function and correlation coefficients were computed by Pearson method.

**Whole exome sequencing and data analysis**. Genomic DNA of ZM-induced primary tumors, derived from bone marrow of leukemic mice and followed by a quick puromycin selection to enrich ZM-expressing leukemic blasts, was extracted using the PureLink Genomic DNA Mini Kit (Invitrogen, cat# K182002). Genomic DNA of the ZM-infected HSPCs, which were used in bone marrow transplantation for inducing AML in mice, was used as control for calling potential spontaneous mutations acquired during the course of AML development in vivo. Library preparation, sequencing, and data analysis were carried out by Novogene. In brief, sequencing libraries were generated using the Agilent SureSelectXT Mouse All Exon kit. After 100X whole exome sequencing, BWA was utilized to map the paired-end clean reads to the mouse reference genome mm10. SNPs and InDels were detected and filtered (total counts ≥ 10) using GATK software. The tool muTect and Strelka were used to call somatic SNPs and InDels in tumor samples, respectively. Variants were annotated with ANNOVAR.

**Quantitative RT-qPCR and ChIP-qPCR**. RT-qPCR and ChIP-qPCR were carried out as described before[3,66]. Briefly, the extracted total RNAs were converted to cDNA with the iScript cDNA Synthesis Kit (Bio-Rad, 1708890). Quantitative PCR was performed in triplicate using the iTaq Universal SYBR Green Supermix (Bio-Rad, 1725124) on an ABI 7900HT fast real-time PCR system. The qPCR signal was first normalized to 18S ribosomal RNA (in RT-qPCR) or input DNA (in ChIP-qPCR) for input normalization, followed by a second normalization to a control cell for calculating relative signals. The detailed primer sequences are provided in the Supplementary Data 6.

**BioID**. The HSPCs stably expressing BirA-tagged ZM were treated with 50uM of biotin for 24 h to allow for proximity-dependent biotinylation of ZM-interacting proteins, followed by lysis in RIPA buffer (10% glycerol, 25 mM Tris-HCl pH 8, 150 mM NaCl, 2 mM EDTA, 0.1% SDS, 1% NP-40, 0.2% Sodium Deoxycholate) freshly supplemented with 1X protease inhibitor cocktail, 1 mM PMSF, and 250 units of Benzonase for 1 h with rotation at 4 °C. After centrifugation at max speed for 30 min at 4 °C, the cleared supernatant was incubated with Neutravidin beads (Thermo Fisher, 29204) overnight at 4 °C. The beads were washed sequentially with 1 mL of RIPA buffer, TAP lysis buffer (10% glycerol, 150 mM NaCl, 2 mM EDTA, 0.1% NP-40, 50 mM HEPES pH 8) and ABC buffer (50 mM Ammonium bicarbonate, pH 8), followed by mass spectrometry-based analysis.

**Mass spectrometry-based protein identification**. Proteins were eluted from beads by heating at 95 °C for 5 min in 100 μL 1x Laemmli buffer (Boston Bioproducts), followed by separation in a 4–12% Bis-Tris Deep Well gel and visualization by Coomassie staining. Each gel lane was collected and cut into 12 equal-volume gel slices, which were subject to in-gel trypsin digestion as described before[80]. In brief, gel segments were destained (in 50 mM ammonium bicarbonate buffer with 50% methanol) first, followed by reduction (in 10 mM TCEP), alkylation (in 50 mM iodoacetamide), dehydration (in acetonitrile) and finally digestion at 37 °C for 12–16 h in 50 mM ammonium bicarbonate buffer containing 100 ng porcine sequencing grade modified trypsin (Promega). Tryptic peptide products were then acidified (in 0.1% formic acid) and separated with an in-line 150 × 0.075 mm column loaded with reverse phase XSelect CSH C18 2.5 um resin (Waters) using a nanoAcquity UPLC system (Waters). Eluted peptides were ionized by electrospray (2.15 kV) and analyzed using an Orbitrap Fusion Tribrid mass spectrometer (Thermo), with MS data and MS/MS data acquired by the FTMS

(profile mode; with resolution of 240,000 and range from 375 to 1500 m/z) and ion trap (centroid mode; with normal mass range and precursor mass-dependent normalized collision energy between 28.0 and 31.0) analyzer, respectively.

**Proteomic data analysis**. Proteins were identified by searching the UniProtKB database restricted to Mus musculus using Mascot (Matrix Science) with a parent ion tolerance of 3 ppm and a fragment ion tolerance of 0.5 Da, fixed modifications for carbamidomethyl of cysteine, and variable modifications for oxidation on methionine and acetyl on N-terminus. Scaffold (Proteome Software) was used to verify MS/MS based peptide and protein identifications. Peptide identifications were accepted if they could be established with less than 1.0% false discovery by the Scaffold Local FDR algorithm. Protein identifications were accepted if they could be established with less than 1.0% false discovery and contained at least 2 identified peptides. Protein probabilities were assigned by the Protein Prophet algorithm[81]. Proteins were filtered if they had a spectral count < 8 in at least one sample group, and the counts were normalized to log2 normalized spectral abundance factor (NSAF) values[82]. Significant interacting proteins were identified by a log2 fold change > 4.

**Compound treatment**. TH1834 (Axon, cat#2339) or I-BET151 (Selleckchem) were dissolved in DMSO and used for cell treatment studies. To perform treatment studies with the leukemia-bearing mice (usually seven days post-BMT), I-BET151 was first dissolved in DMSO to make a stock solution of 120 mg/mL, followed by dilution to 6 mg/mL by mixing 50uL of the I-BET151 stock with 950uL of vehicle solution (20% Kolliphor HS 15 [v/v] in 0.9% of NaCl; Sigma, 42966). I-BET151 or mock vehicle control was delivered via intraperitoneal (i.p.) injection at a daily dose of 30 mg/kg body weight for three weeks (5 days' injections per week).

**Statistics and reproducibility**. Data in bar and line charts are presented as mean ± SD of three independent experiments unless otherwise noted. Statistical analysis was performed with two-sided Student's t test for comparing two sets of data with assumed normal distribution. We used a two-sided log-rank test for Kaplan-Meier survival curves to determine statistical significance. A $p$ value of less than 0.05 was considered significant. Statistical significance levels are denoted as follows: *$p < 0.05$; **$p < 0.01$; ***$p < 0.001$; ****$p < 0.0001$. No statistical methods were used to predetermine sample size. All data from representative experiments (such as western blot image and other micrographs) were repeated two to three times independently with similar results.

**Reporting summary**. Further information on research design is available in the Nature Research Reporting Summary linked to this article.

## Data availability

RNA-seq, ChIP-seq and CUT&RUN datasets related to this work have been deposited in the NCBI GEO under accession number GSE150428. The mWES data are available in the NCBI SRA under the accession code PRJNA693299. The Mass Spectrometry proteomics data have been deposited to the ProteomeXchange Consortium via the PRIDE partner repository with the dataset identifier PXD023702. The source data underlying Figs. 1r, 5b, c, h, 6b, e–g and Supplementary Figs. 5a–c, 7a are provided in the Source Data file. All the other data supporting the findings of this study are available within the article and its Supplementary Information files and from the corresponding author upon reasonable request. Source data are provided with this paper.

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

## Acknowledgements

We graciously thank BD Strahl, R Rottapel, MP Kamps and Q Zhang for providing reagents used in the study and all members of the Wang Laboratory for helpful discussion and technical supports. We thank UNC core facilities, including High-throughput Sequencing Facility, Bioinformatics Core, Flow Cytometry Core, Animal Studies Core and Histology Research Core, for their professional support of this work. The cores affiliated to UNC Cancer Center are supported in part by the UNC Lineberger Comprehensive Cancer Center Core Support Grant P30-CA016086. This work utilized the AVANCE NEO 600 MHz NMR Spectrometer System that was upgraded with funding from a NIH SIG grant 1S10OD025132-01A1. We acknowledge proteomics support from the NIH IDeA National Resource for Quantitative Proteomics with funding through TL1TR003109, P20GM121293, R24GM137786, S10OD018445, and R01CA236209. This work was supported by NIH grants (R01-CA215284 and R01-CA211336 to G.G.W., R01GM122749 to J.J.), and grants of a Concern Foundation for Cancer Research grant (to G.G.W.), Gabrielle's Angel Foundation for Cancer Research (to G.G.W.), Gilead Sciences Research Scholars Program in hematology/oncology (to G.G.W.) and When Everyone Survives (WES) Leukemia Research Foundation (to G.G.W.). G.G.W. is an American Cancer Society (ACS) Research Scholar, an American Society of Hematology (ASH) Scholar in basic science and a Leukemia and Lymphoma Society (LLS) Scholar.

## Author contributions

J.L. designed the research, performed experiments, interpreted data and wrote the manuscript. P.G., J.L., W.G., Y.T., Y.G., and L.C. analyzed genomic datasets under the supervision of D.Z. and G.G.W.. A.J.S., S.G.M., R.D.E., and S.D.B. performed proteomics studies under the supervision of A.J.T.. X.Y. and J.J. synthesized compound. S.H., J.H.A. and Y.G. contributed to cell sorting, RNA-seq library generation and plasmid construction, respectively. J.E.F. surveyed publicly available cancer patient datasets. G.G.W. conceived the idea, supervised and designed the research, interpreted data, and wrote the manuscript with the inputs from all authors.

## Competing interests
