## [Peer Review File · Nature Communications]

REVIEWER COMMENTS

Reviewer #1 (Remarks to the Author):

Li and colleagues evaluate the transformation potential of a rare fusion described in a handful of AML cases, ZMYND11-MBTD1. No functional data to date exists on this fusion, and although this is a rare fusion the mechanism of leukemogenesis has broader implications and is therefore important and warrants publication. The authors demonstrate this fusion blocks differentiation, maintains a stem cell like gene expression program, and efficiently transforms hematopoietic cells. Further they map genome occupancy and show co-localization with H3K36me3 and H3K27ac marks as well as TIP60 in addition to demonstrating that TIP60 directly binds to/interacts with the fusion. Several mutant fusion constructs were generated to abolish chromatin association and TIP60 association and assess their impact on the fusion. Overall the manuscript is extremely well written, the experiments are extensive and provide a very comprehensive overview of how this fusion functions.

Major Comments:

The data evaluating the transformation capabilities of ZMYND11 and MBTD1 mutants are limited to in vitro assays. The major ones required for chromatin and TIP60 association should be introduced in a transplantation assay similar to figure 1h to show the significance in vivo.

I was unable to locate the normalization factor used for ChIPseq analysis in the methods. The normalized read counts for some of the data appear low - in particular Brd4 at the Meis1 locus(0-3; fig 7b). How robust are these peaks. We typically use a normalization factor of 15 so a value of 3 means 0.2 counts per million.

Minor Comments:

Figures 4c and 7a heatmaps appear to show the four clusters seen in Fig 3e. The cluster designation on the y axis of the heatmaps are missing.

Reviewer #2 (Remarks to the Author):

In this paper, the authors explore the role of the ZMYND11-MBTD1 (ZM) fusion in leukemogenesis. They show that a ZM construct can immortalize primary mouse bone marrow cells and cause leukemia (with some latency) in mice. Inclusion of an Nras mutation with the ZM fusion produces a leukemia with a much reduced latency, suggesting a cooperation between these two mutations. They perform RNA-seq and using a "stemness" signature to show that ZM expresses the Hoxa cluster and other genes associated with the LSK compartment, and that it has a signature somewhat reminiscent of MLL-AF9 leukemias. ZM ChIP-seq revealed a pattern that anti-correlates with H3K727me3 and correlates with nearby enrichment for H3K36me3 in the associated bound genes. Using BioID, they identify Tip60 (and other proteins) as a ZM co-purifying factor, that also strongly correlates with ZM in ChIP-seq assays, especially at "superenhancers". Mutations that abolish this interaction abolish the ability of ZM to immortalize BM cells. They then show that the ZMYND11 domains function as a site recruitment module, mainly through Bromo-PWWP recognition of H3K36me3, and this is also essential for immortalization. ZM binding is also associated with Brd4, and treatment of mouse leukemias with the Brd4 inhibitor I-BET151 abolished leukemic growth and increased survival.

Overall, the experiments are extremely well done and the data is very convincing. It is also a very comprehensive and complete study. I just have a few points and questions.

1. Is there any evidence of a co-occurrence between ZM fusions and Ras mutations (or any other associated mutations) in patients? This would help provide evidence that the two hit model proposed here is representative of the human disease.
2. Are there any spontaneous mutations in the ZM alone mouse leukemias? This could explain why some of the primary recipients did not seem to get leukemia.
3. Lin⁻ cells were used as the transformation target(s) for the ZM construct, but since there are many different cell types present it is hard to know what the target cell type for transformation is. For example, can ZM fusions also transform purified GMP cells or does it specifically require HSCs?
4. Among the examples of ZM superenhancers shown in Figure 4, the only intergenic enhancer is in 4i. The other examples show promoter binding and spreading into the gene, which could be an association with transcription elongation rather than enhancer activity. It would be useful to determine how many of the identified superenhancers in Figure 4 are over a gene body, and how many are intergenic. It would also be nice to see some other tracks of intergenic examples.
5. ZM seems to primarily bind to H3K36me3, but if you look at the CHIP-seq heat maps and example tracks in Figure 3 e-f, it looks more like ZM binds primarily next to sites of H3K36me3 enrichment, and generally doesn't spread across H3K36me3 domains (e.g. see Meis1 in Figure 3f). How do the authors explain this? And what is recruiting ZM to the H3K36me3 non-enriched regions (e.g. several places in the Hox loci, the Meis1 promoter)?
6. Brd4 inhibition has something of a generic effect across most AMLs (including MLL-AF9 leukemias) probably due to the generally important role this co-activator has in the activation of multiple genes, including MYC. However, despite the success of these inhibitors in mouse models, this class of inhibitor has performed fairly poorly in clinical trials. With this in mind, how likely do the authors think that their results here will be transferable to human patients?

General Response to Reviewers

First, we appreciate the general enthusiasms from reviewers about our work, such as

(Reviewer 1) “the paper *has broader implications and is therefore important and warrants publication*” and that “*Overall the manuscript is extremely well written, the experiments are extensive and provide a very comprehensive overview of how this fusion functions*”.

(Reviewer 2) “*Overall, the experiments are extremely well done and the data is very convincing. It is also a very comprehensive and complete study.*”

Also, we wanted to thank the reviewers for providing us the critical comments and constructive suggestions on how to further improve our manuscript. We have made every effort to carry out extensive experiments to address all of the comments and concerns from reviewers. In particular, we have conducted these following **major** experiments to strengthen our findings:

- 1) We compared the AML-inducing capabilities of WT ZM with that of two critical mutants with a defect in either stable chromatin association (ZM^{W294A}) or Tip60 interaction (ZM^{ALL>DDD}) using mouse model in the presence of Nras^{G12D}.
- 2) We performed whole exome sequencing to identify spontaneous gene mutations among the ZM-induced primary murine AMLs. In one AML sample, we identified a gain-of-function/activating mutation S502L in Ptpn11/Shp2, which was previously reported to be an activator of RAS signaling.
- 3) We carried out CUT&RUN to map out and compare the genome-wide binding patterns of different versions of ZM, either WT or with its ZMYND11 or MBTD1 segment deleted (see new data of **Fig. 6i-j and Supplementary Fig. 7c-e**). This is to determine how the ZMYND11 and MBTD1 segments contribute to ZM's chromatin targeting.

We thank the reviewers for their helpful suggestions, with which the manuscript has been significantly improved, and hope that our manuscript is now acceptable for publication in *Nature Communications*.

Point-by-point Response

Response to Reviewer 1

Li and colleagues evaluate the transformation potential of a rare fusion described in a handful of AML cases, ZMYND11-MBTD1. No functional data to date exists on this fusion, and although this is a rare fusion the mechanism of leukemogenesis has broader implications and is therefore important and warrants publication. The authors demonstrate this fusion blocks differentiation, maintains a stem cell like gene expression program, and efficiently transforms hematopoietic cells. Further they map genome occupancy and show co-localization with H3K36me3 and H3K27ac marks as well as TIP60 in addition to demonstrating that TIP60 directly binds to/interacts with the fusion. Several mutant fusion constructs were generated to abolish chromatin association and TIP60 association and assess their impact on the fusion. Overall the manuscript is extremely well written, the experiments are extensive and provide a very comprehensive overview of how this fusion functions.

We appreciate the reviewer's positive comments that our finding “*has broader implications and is therefore important and warrants publication*” and that “*Overall the manuscript is extremely well written, the experiments are extensive and provide a very comprehensive overview of how this*

fusion functions". We also thank the reviewer for his/her critical comments as listed in below sections.

Main critique 1: *The data evaluating the transformation capabilities of ZMYND11 and MBTD1 mutants are limited to in vitro assays. The major ones required for chromatin and TIP60 association should be introduced in a transplantation assay similar to figure 1h to show the significance in vivo.*

Response:

We appreciate the reviewer's comment. Following the above suggestion, we have used a Nras co-expression system used in *figure 1h*, and as shown in the below **Fig R1**, the co-expression of WT ZM plus Nras rapidly induced AML in mice with an average latency of 42 days whereas co-expression of Nras with a mutant ZM, defective in either chromatin association (W294A in *ZMYND11* part) or Tip60 interaction (ALL>DDD in *MBTD1* part), has not induced AMLs during the same monitoring period in recipient mice. Meanwhile, our bioluminescence imaging of live animals in the latter mutant cohort showed a significant expansion of virally infected HSPCs in bone marrow and spleen over the monitoring period. Difference of AML kinetics in this in vivo assay is significant (P=0.004) and in agreement with our repeated in vitro assays showing that the mutant ZM^{W294A} or ZM^{ALL>DDD} was unable to transform HSPCs. In addition, per request of Reviewer 2 (see below), we have carried out CUT&RUN to map out the genome-wide binding patterns of different forms of ZM, either WT or with its *ZMYND11* or *MBTD1* segment deleted, and the results show that ZM's chromatin binding locations are determined by both ZMYND11 and MBTD1 segments, with the former one (ZMYND11) being the major determinant.

Together, these in vivo, in vitro and genomics results provide a strong support that intact *ZMYND11* and *MBTD1* segments mediate *chromatin binding and Tip60 association*, both of which are required for inducing AML.

Fig R1. Either of the two mutant ZM forms, with a defect in either chromatin association (W294A) or Tip60 interaction (ALL>DDD), failed to rapidly induce murine AMLs in the presence of Nras^{G12D}.

a, Kaplan-Meier survival curve of syngeneic mice post-transplantation of HSPCs stably transduced by the indicated genes (n=cohort size). The P value was calculated by log-rank test.

b, Bioluminescence images of mice at different time points post BM transplantation.

Main critique 2. *I was unable to locate the normalization factor used for ChIPseq analysis in the methods. The normalized read counts for some of the data appear low - in particular Brd4 at the Meis1 locus (0-3; fig 7b). How robust are these peaks. We typically use a normalization factor of 15 so a value of 3 means 0.2 counts per million.*

Response:

We thank Review #1 for raising this concern. We did a two-step normalization when generating the bigwig coverage file from the BAM alignment file: the number of reads per bin was first normalized to 1x genome coverage (**reads per genome coverage, RPGC**) and then normalized to input by subtract using deepTools bamCoverage and bigwigCompare functions, respectively. We didn't use additional scale factor to artificially increase the normalized counts, as the reviewer indicated. So, a value of 3 in our figure represents 3 read counts **per 1x genome coverage (RPGC)** after subtracting the corresponding counts in that bin in the input sample. We believe that the robustness of the Brd4 peaks are comparable to what was shown in the published works. For example, there are 2 narrow peaks being called at intron 2 of Meis1 with a $-\log_{10}p$ value of 7.55 and 4.70 respectively (please check the below **Fig R2**).

We are sorry for this confusion. We have provided more details regarding the used scale in the revised figure legends and also the normalization steps in the revised Methods section.

Fig R2. Robustness of the Brd4 peaks demonstrated by 2 narrow peaks being called at intron 2 of Meis1 (grey lines), with a $-\log_{10}p$ value of 7.55 and 4.70 respectively.

Minor Comments:

1. Figures 4c and 7a heatmaps appear to show the four clusters seen in Fig 3e. The cluster designation on the y axis of the heatmaps are missing.

Response:

Thank reviewer for pointing this out, and we have added the missing cluster labels.

Response to Reviewer 2

In this paper, the authors explore the role of the ZMYND11-MBTD1 (ZM) fusion in leukemogenesis. They show that a ZM construct can immortalize primary mouse bone marrow cells and cause leukemia (with some latency) in mice. Inclusion of an Nras mutation with the ZM fusion produces a leukemia with a much reduced latency, suggesting a cooperation between these two mutations. They perform RNA-seq and using a “stemness” signature to show that ZM expresses the Hoxa cluster and other genes associated with the LSK compartment, and that it has a signature somewhat reminiscent of MLL-AF9 leukemias. ZM ChIP-seq revealed a pattern that anti-correlates with H3K727me3 and correlates with nearby enrichment for H3K36me3 in the associated bound genes. Using BioID, they identify Tip60 (and other proteins) as a ZM co-purifying factor, that also strongly correlates with ZM in ChIP-seq assays, especially at “superenhancers”. Mutations that abolish this interaction abolish the ability of ZM to immortalize BM cells. They then show that the ZMYND11 domains function as a site recruitment module, mainly through Bromo-PWWP recognition of H3K36me3, and this is also essential for immortalization. ZM binding is also associated with Brd4, and treatment of mouse leukemias with the Brd4 inhibitor I-BET151 abolished leukemic growth and increased survival.

Overall, the experiments are extremely well done and the data is very convincing. It is also a very comprehensive and complete study. I just have a few points and questions.

We thank Reviewer #2 for his/her positive comment on our work and please see below sections for our responses to the raised issues.

1. Is there any evidence of a co-occurrence between ZM fusions and Ras mutations (or any other associated mutations) in patients? This would help provide evidence that the two hit model proposed here is representative of the human disease.

Response:

Reviewer #2 raised an important question as for “*co-occurrence between ZM fusions and Ras mutations in patients*”. To address the issue, we established a collaboration with Dr. Jason E Farrar, an AML clinician and investigator involved in the NIH-funded “TARGET Initiative” focused on childhood cancers that cover hundreds of AML patients treated nationwide through the Children’s Oncology Group. We found that ~30% of these pediatric AML patients carry an N-RAS mutation and the RAS mutation percentage is closer to 40-50%, if we add in K-RAS (almost no mutation of H-RAS). Meanwhile, ZM fusion is rather rare. In order to calculate the significance for “*co-occurrence between two*”, we need a much larger size of patients, which are unavailable to us to pursue further at the moment. Furthermore, as requested in the below comment, we have performed the whole Exome sequencing of primary samples of murine AMLs caused by ZM alone and found that one AML carries high rate of missense mutation of S502L in Ptpn11 (aka, Shp2). PTPN11 is an activator of RAS/MAPK cascade, and what is amazing is that, the same S502L mutation of PTPN11/SHP2 was previously identified from human leukemia patients, leading to sustained RAS activation (please see below for details). Thus, the functional cooperation study between RAS and ZM is appropriate and relevant.

2. Are there any spontaneous mutations in the ZM alone mouse leukemias? This could explain why some of the primary recipients did not seem to get leukemia.

Response:

The reviewer’s comment is well taken. To determine potential spontaneous mutation that occurs during ZM-induced leukemogenesis in vivo, we did mWES (mouse whole exome sequencing) with two primary mouse AML tumor samples. Here, the originally transplanted HSPC pool (ZM-infected) was used as a control to call somatic mutations during the in vivo clonal evolution. We indeed found the existence of a few acquired nonsynonymous, exonic mutations (please check the new **Supplementary Table 1**).

To our excitement, Ptpn11 (Tyrosine-protein phosphatase non-receptor type 11; also known as Shp2) is mutated in one tumor, with missense C1505T mutation of Ptpn11 causing the Ser 502 to Leu substitution (S502L). Ser502, located at the catalytic PTP (protein tyrosine phosphatase) domain, is conserved across species such as mouse and human (please check the revised **Supplementary Fig. 2g**). Importantly, S502L and a similar S502P mutation of PTPN11/SHP2 were reported among human patients with Noonan syndrome and/or leukemia, and both mutations were predicted to be pathogenic likely through destabilizing the auto-inhibited conformation of PTPN11/SHP2, thereby keeping the enzyme in an open, active state and eventually leading to sustained activation of RAS/MAPK pathway. As such, the spontaneous acquisition of an activating mutation in Ptpn11 in ZM alone-initiated AML suggests there is an oncogenic collaboration between ZM and a hyper-activated RAS/MAPK cascade. This finding also provides a rationale to co-express ZM and Nras^{G12D} in our in vivo assays, in which the mutant Nras significantly accelerated the disease progression.

Of note, mutations of Sntg1 and Clca3a2 exist in the other examined AML tumor, implying the presence of other cooperating oncogenic pathways, in addition to Ptpn11/Ras. Further

research is warranted to study whether and how the mutant Sntg1 and Clca3a2 contribute to AML tumorigenesis.

We thank the reviewer for bringing up this question. The new results strengthen our findings and support the requirement of a second, cooperating mutation for ZM to cause full-blown AML.

3. Lin- cells were used as the transformation target(s) for the ZM construct, but since there are many different cell types present it is hard to know what the target cell type for transformation is. For example, can ZM fusions also transform purified GMP cells or does it specifically require HSCs?

Response:

The reviewer has raised an interesting point, as AML may develop from different cellular origins, which may considerably influence the clinical outcome and therapy response. To sort out what cell type is susceptible to transformation by ZM, we used flow-based sorting of mouse bone marrow cells and purified the below three cell types:

primitive HSC (Lin-/cKit+/Sca1+/Cd16/32-/Cd150+/Cd48-),
more committed GMP (Lin-/cKit+/Sca1-/Cd16/32+/Cd150-),
and differentiated myeloid cells (cKit-/Mac1+).

Following ZM transduction, we found that ZM expression efficiently confers both HSC and GMP, but not differentiated myeloid cells, the capability of long-term in vitro proliferation. HSC-derived leukemia cells proliferated slightly faster than those derived from GMP. Moreover, both ZM-immortalized lines exhibit the typical myeloid blast immunophenotype (cKit+/Cd34+/Mac1+ by FACS) and blast morphology (as revealed by W&G staining). These new data are included in the revised **Supplementary Fig. 1**. Overall, we show that ZM can transform both HSC and GMP, but not those differentiated myeloid cells.

4. Among the examples of ZM super-enhancers shown in Figure 4, the only intergenic enhancer is in 4i. The other examples show promoter binding and spreading into the gene, which could be an association with transcription elongation rather than enhancer activity. It would be useful to determine how many of the identified superenhancers in Figure 4 are over a gene body, and how many are intergenic. It would also be nice to see some other tracks of intergenic examples.

Response:

We understand the reviewer's question. Based on our annotation of the superenhancers (SEs) called by either H3K27ac, ZM or Tip60 ChIP signals, we found that SEs in all cases are indeed mostly enriched at introns and then at intergenic regions (see below **Fig R3**). This is consistent with prior reports showing that mammalian SEs tend to overlap with gene bodies¹. Please also refer to additional examples of ZM-activated genes with intergenic SEs in the below **Fig R3**.

Fig R3. Analysis of super-enhancers identified in ZM-induced AML cells.

a, distribution of SEs across different genomic elements.

b-d, IGV views for ZM-activated genes with intergenic SEs.

5. ZM seems to primarily bind to H3K36me3, but if you look at the ChIP-seq heat maps and example tracks in Figure 3 e-f, it looks more like ZM binds primarily next to sites of H3K36me3 enrichment, and generally doesn't spread across H3K36me3 domains (e.g. see Meis1 in Figure 3f). How do the authors explain this? And what is recruiting ZM to the H3K36me3 non-enriched regions (e.g. several places in the Hox loci, the Meis1 promoter)?

Response:

We thank the reviewer for raising this very interesting point. Given that ZM is a fusion protein and exerts its oncogenic functions partly by interacting with the NuA4/TIP60 complex, we speculated that its chromatin targeting location is influenced by both fusion segments. To directly test this idea, we employed the CUT&RUN technique to profile genome-wide bindings of full-length ZM, ZM Δ M (ZM with the MBTD1 part deleted) and ZM Δ Z (ZM with the ZMYND11 part deleted) in 293T cells. CUT&RUN was carried out with two replicates using either Flag or HA antibody, which are highly correlated (please see the new **Supplementary Fig. 7c**).

In essence, ZM binds mainly to both TSS-proximal regions and gene bodies. Deletion of the MBTD1 part (ZM Δ M) causes significant shifting of ZM binding towards gene body, suggesting that the MBTD1 segment confers ZM the ability to target TSS regions (please see the new **Fig. 6i-j and Supplementary Fig. 7d-e**). In contrast, ZM Δ Z is relocated almost exclusively to TSS regions, suggesting that the ZMYND11 part mediates ZM binding to gene body (please see new **Fig. 6i-j and Supplementary Fig. 7d-e**). Notably, although ZM Δ M is more enriched in gene body compared to full-length ZM, a subset of its peaks are also present at TSSs, suggesting ZMYND11

part harbors unknown TSS-targeting activity (for example, WT ZMYND11 was suggested before to interact with certain transcription factors). Furthermore, the Pearson correlation analysis among all samples revealed a closer similarity between binding pattern of full-length ZM and that of ZMΔM, relative to ZMΔZ, suggesting that the ZMYND11 segment plays a major role in chromatin targeting of ZM (please see the new **Supplementary Fig. 7c**).

Together, our new data demonstrate that both ZMYND11 and MBTD1 segments modulate the genomic binding locations of ZM, with the former one being the major determinant, which explains that “*ZM binds primarily next to sites of H3K36me3 enrichment, and generally doesn't spread across H3K36me3 domains*”. We have added a new section in the revised main text to cover these results. We thank the reviewer's comment, which helped improve the paper.

6. Brd4 inhibition has something of a generic effect across most AMLs (including MLL-AF9 leukemias) probably due to the generally important role this co-activator has in the activation of multiple genes, including MYC. However, despite the success of these inhibitors in mouse models, this class of inhibitor has performed fairly poorly in clinical trials. With this in mind, how likely do the authors think that their results here will be transferable to human patients?

Response:

We appreciate this thought on the translational aspect. As the reviewer correctly pointed out, the currently available BET inhibitors as single agents seem to only exhibit limited efficacy in clinical trials, although bromodomain inhibitors show highly promising effects among various mouse models (as shown by this work and many other published papers). Part of the reasons that bromodomain inhibitors performed poorly in clinical trials seems to be a lack of sufficient potency and/or existence of unwanted toxicity. Currently, researchers are actively developing next-generation BET-inhibiting compounds, such as bivalent BET inhibitors which bind the two tandem bromodomains (BD1 and BD2) simultaneously^{2, 3}, BET degraders based on the PROTAC technology⁴, as well as bromodomain-specific ones which selectively target BD1 or BD2 in BET proteins⁵. Studies of these new agents in the pre-clinical setting show an early promise in offering either superior therapeutic potency or improved specificity over traditional inhibitors. We also favor a view that efficacy of BET inhibition can be further boosted through combination therapies with other synergistic anticancer drugs. Overall, we remain to be optimistic and look forward to further advances along these lines in years to come. We have added a brief discussion of this point at Discussion.

Reference used in the Response Letter

1. Perez-Rico YA, *et al.* Comparative analyses of super-enhancers reveal conserved elements in vertebrate genomes. *Genome Res* **27**, 259-268 (2017).
2. Ren C, *et al.* Spatially constrained tandem bromodomain inhibition bolsters sustained repression of BRD4 transcriptional activity for TNBC cell growth. *Proc Natl Acad Sci U S A* **115**, 7949-7954 (2018).
3. Rhyasen GW, *et al.* AZD5153: A Novel Bivalent BET Bromodomain Inhibitor Highly Active against Hematologic Malignancies. *Mol Cancer Ther* **15**, 2563-2574 (2016).
4. Winter GE, *et al.* DRUG DEVELOPMENT. Phthalimide conjugation as a strategy for in vivo target protein degradation. *Science* **348**, 1376-1381 (2015).

5. Gilan O, *et al.* Selective targeting of BD1 and BD2 of the BET proteins in cancer and immunoinflammation. *Science* **368**, 387-394 (2020).

REVIEWERS' COMMENTS

Reviewer #1 (Remarks to the Author):

The authors have done a significant amount of work to address the reviewers concerns, strengthening the manuscript. I have no further comments. It is a very nice manuscript.

Reviewer #2 (Remarks to the Author):

The authors have done an amazing job of answering my questions and concerns. The new work has added some surprising and exciting new results to the paper and has increased how interesting the final paper is. I think this is an excellent and very well performed study.

Point-by-point Response

REVIEWERS' COMMENTS

Reviewer #1 (Remarks to the Author):

The authors have done a significant amount of work to address the reviewers concerns, strengthening the manuscript. I have no further comments. It is a very nice manuscript.

Response: We appreciate reviewer 1's positive views.

Reviewer #2 (Remarks to the Author):

The authors have done an amazing job of answering my questions and concerns. The new work has added some surprising and exciting new results to the paper and has increased how interesting the final paper is. I think this is an excellent and very well performed study.

Response: We appreciate reviewer 2's recommendation for publication of this paper.